# Long-term safety and effectiveness of mRNA-1273 vaccine in adults: COVE trial open-label and booster phases

Primary vaccination with mRNA-1273 (100-µg) was safe and efficacious at preventing coronavirus disease 2019 (COVID-19) in the previously reported, blinded Part A of the phase 3 Coronavirus Efficacy (COVE; NCT04470427) trial in adults (≥18 years) across 99 U.S. sites. The open-label (Parts B and C) primary objectives were evaluation of long-term safety and effectiveness of primary vaccination plus a 50-µg booster dose; immunogenicity was a secondary objective. Of 29,035 open-label participants, 19,609 received boosters (mRNA-1273 [n = 9647]; placebo-mRNA-1273 [n = 9952]; placebo [n = 10] groups). Booster safety was consistent with that reported for primary vaccination. Incidences of COVID-19 and severe COVID-19 were higher during the Omicron BA.1 than Delta variant waves and boosting versus non-boosting was associated with a significant, 47.0% (95% CI : 39.0-53.9%) reduction of Omicron BA.1 incidence (24.6 [23.4 – 25.8] vs 46.4 [40.6 – 52.7]/1000 person-months). In an exploratory Cox regression model adjusted for time-varying covariates, a longer median interval between primary vaccination and boosting (mRNA-1273 [13 months] vs placebo-mRNA-1273 [8 months]) was associated with significantly lower, COVID-19 risk (24.0% [16.0% – 32.0%]) during Omicron BA.1 predominance. Boosting elicited greater immune responses against SARS-CoV-2 than primary vaccination, irrespective of prior SARS-CoV-2 infection. Primary vaccination and boosting with mRNA-1273 demonstrated acceptable safety, effectiveness and immunogenicity against COVID-19, including emergent variants.

The phase 3 Coronavirus Efficacy (COVE; NCT04470427) trial demonstrated the safety and efficacy of the mRNA-1273 vaccine in preventing SARS-CoV-2 infection in adults, including severe disease[1,2]. The mRNA-1273 vaccine two-injection primary series demonstrated 93.2% efficacy against coronavirus disease 2019 (COVID-19) and acceptable safety at the completion of the blinded, randomized, placebo-controlled Part A of the COVE trial in adults with a median follow-up time of 5.3 months[1]. The vaccine has been approved by many regulatory authorities and has been widely administered globally to children and adults[3–6]. Real-world

effectiveness was also demonstrated against COVID-19, especially against severe disease and death[7–11].

The mRNA-1273 vaccine exhibited robust antibody responses across age groups, neutralized severe acute respiratory syndrome coronavirus-2 (SARS-CoV-2) variants, and was effective against SARS-CoV-2 variants circulating during the blinded and open-label periods of the COVE trial[1,2,7,10,12–18]. An additional 50-µg mRNA-1273 vaccine dose was authorized as a first booster dose and has demonstrated improved immune responses and effectiveness against variants[16,19–22]. As variants became more divergent, such as the Omicron family of variants,

✉ e-mail: lbaden@bwh.harvard.edu

updated versions of mRNA-1273 bivalent and monovalent-containing variant spike mRNAs[23–27] were authorized and/or approved in multiple regions[28,29].

Following the Emergency Use Authorization (EUA) for COVID-19 vaccines in December 2020, the COVE protocol was amended to include an open-label component (Part B), offering participants the option to unblind and for placebo recipients to receive mRNA-1273. The protocol was further amended to offer a 50-µg booster dose of mRNA-1273 to coincide with its authorization in October 2021[30,31], to all vaccinated participants in the study (Part C; Fig S1). Herein, the final results of the open-label and booster Parts (B and C) of the trial through study-end (April 2023) are reported.

## Results
### Trial population
A total of 29,035 participants started open-label Part B, including those who had received mRNA-1273 (mRNA-1273 group) or placebo in Part A and crossed-over (placebo-mRNA-1273 group) to receive at least 1 injection of mRNA-1273 in Part B, and placebo participants who continued the study and did not receive mRNA-1273 in Part B (Fig. 1). In Part C, 9647 participants in the mRNA-1273 and 9952 in the cross-over placebo-mRNA-1273 groups received the booster. Safety was assessed in 30,346 participants for the primary vaccine series and 19,609 participants for the booster, and effectiveness in 28,463 and 16,368 participants, respectively (Fig. S1). Immunogenicity was evaluated in 731 participants in the mRNA-1273 group in the per-protocol immunogenicity set (PPIS) (Figs. S1, S2).

Baseline characteristics of the Part C booster participants were generally balanced and similar to those of the blinded, Part A COVE study cohort[1,2] (Table S1). Prior SARS-CoV-2 infection was present in 198 (2.1%) and 184 (1.8%) of participants at baseline in Part A, and 941 (9.8%) and 1647 (16.5%) at pre-booster in the mRNA-1273 and placebo-mRNA-1273 groups, respectively. Median follow-up times (interquartile ranges [IQRs]) were 392 (373–413) and 250 (236–266) days between the second injection of the primary series to the booster and 333 (309–357) and 325 (299–351) days post-booster follow-up to study-end for the mRNA-1273 and placebo-mRNA-1273 groups, respectively.

### Safety
The median follow-up measured in days (IQR) for the primary series between the first injection of the primary series and the booster administration date, or last date of study participation or study-end, whichever was earliest, was 415 (386–442) in the mRNA-1273 and 281 (266–302) in the placebo-mRNA-1273 groups. Safety was similar to that previously reported for the blinded part of the study[1,2] and post-authorization data with no new safety concerns identified in either group (Tables S2–S4).

Among the 19,609 booster recipients, 34.2% of participants in the mRNA-1273 group and 31.6% in the placebo-mRNA-1273 group experienced adverse events (AEs) ≤ 28 days after the booster dose (Table S5). Of these participants, 25.1% in the mRNA-1273 group and 21.8% in the placebo-mRNA-1273 group had AEs considered by the investigator to be vaccine-related; most were consistent with the known reactogenicity of the vaccine (Table S6). Serious AEs considered by the investigator to be vaccine-related occurred in 3 [<0.1%] participants in the mRNA-1273 group, including a case of myocarditis that occurred on day 1 post-booster that resolved at day 72 in a

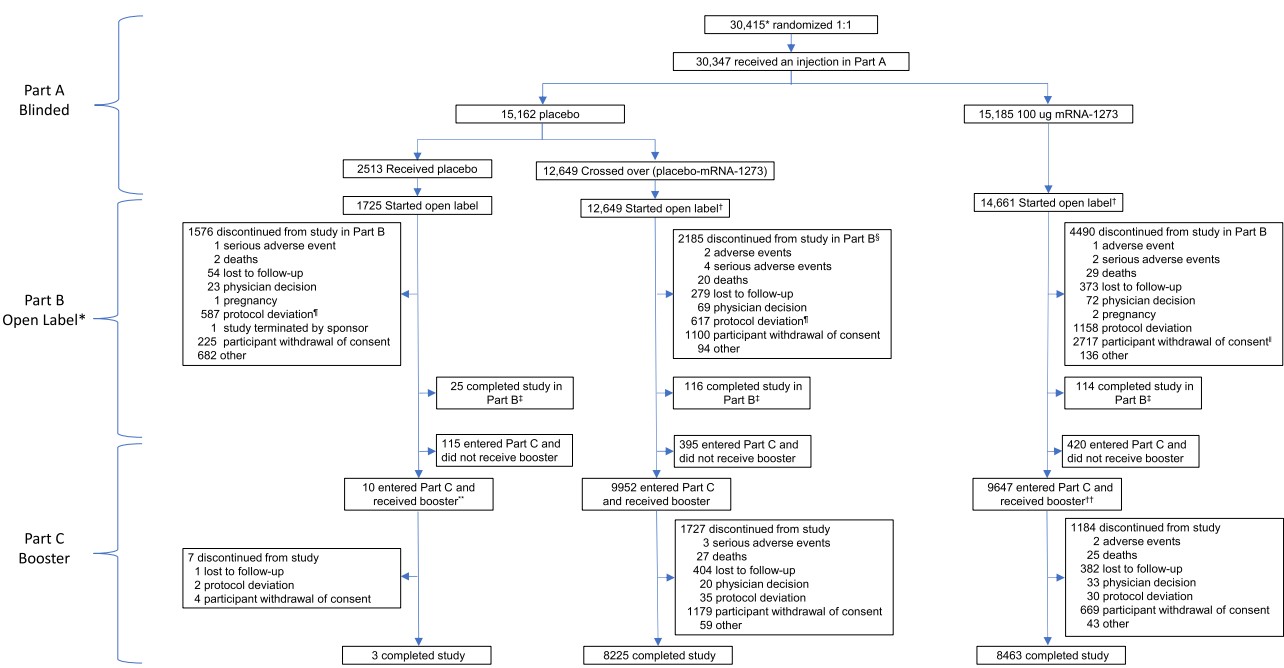

**Fig. 1 | Trial profile of participants in the Parts B and C safety sets.** BDV booster decision visit, FPFV first participant first visit, PDV participant decision visit. *Participants (n = 29,035) who started the open-label observational phase include those who had a PDV or unblinding date. †In Part B, 12,554 of the 12,649 participants in the placebo-mRNA-1273 group who received a first injection of mRNA-1273 also received a second dose. Of the 15,185 participants in the mRNA-1273 group who received at least one injection of mRNA-1273 in Part A, 139 received one injection of mRNA-1273 in Part B. §One participant discontinued after the discontinuation cut-off date. ¶A greater proportion of participants in the placebo compared with the mRNA-1273 group discontinued the study due to protocol deviations, primarily receipt of an off-study COVID-19 vaccine. ‖The higher number of discontinuations by the withdrawal of consent in the mRNA-1273 group is explained largely by the recruitment of participants to other booster dose clinical studies (~3863 participants to phase 2 [NCT04405076] and phase 2/3 [NCT05249829] studies). ‡Participants were considered to have completed the study if they completed the final visit on day 759 (month 25), 24 months following the last injection of study vaccination. **Included one participant who did not enter the open-label observational phase. ††Included 10 participants who did not enter the open-label observational phase. Study initiation dates: for Part A blinded FPFV July 27, 2020; for Part B (PDV) December 2020; Part C (BDV) September 23, 2021. Data-cutoff date: Part A March 26, 2021; Part B Booster day 1 visit or database lock date, April 7, 2023, whichever is earlier; database lock date: Part C April 7, 2023.

participant aged >40 years who had just completed treatment for an upper respiratory infection 1 month prior; a case of coronary arteriospasm with onset on day 11 post-booster in a participant >70 years of age with known coronary artery disease; and a case of "heart flutter" (not verified by electrocardiogram) with onset on day 12 post-booster in a participant >40 years of age being treated for anxiety that resolved without treatment the following day (Table S7). One (<0.1%) serious AE of erythema nodosum occurred in a participant in the placebo-mRNA-1273 group on day 9 post-booster that resolved without treatment on day 30. Fatal AEs were experienced by 6 participants in the booster part of the study; none were considered by the investigator to be vaccine-related. Incidences of AEs ≤28 days after the booster, including those considered by the investigator to be related to study vaccine, were generally similar regardless of pre-booster SARS-CoV-2-infection status, sex, race, and ethnicity (Table S8–S11). As expected, the frequency of SAEs was higher in older participants (≥65 years) compared to younger (≥18–< 65 years) participants (Table S12).

AEs that occurred after the booster dose and up to study-end were reported in 65.4% of all participants, predominantly attributable to underlying illness or intercurrent infection or injury, including 23.6% events considered by the investigator to be related to the study vaccine (Table S13). No additional SAEs considered by the investigator to be related to study vaccination occurred after 28 days. A total of 50 (0.3%) fatal AEs occurred as of study-end; none were attributed to study vaccine. Four deaths were associated with COVID-19, two during Part B (COVID-19 and pneumonia bacterial, and acute respiratory failure) and two after booster in Part C (respiratory failure [updated to COVID-19 after database lock] and COVID-19).

### Effectiveness

Among 1055 adjudicated COVID-19 cases that occurred in the study prior to the booster dose, Delta (46.1%) and Omicron BA.1 (16.2%) were the most frequently detected variants (Table S14). Among 3538 COVID-19 cases that occurred following the booster dose, Omicron variants were the most frequently detected variants (75.0%), including subvariants BA.1 (1109 [31.3%], BA.2 (744 [21.0%]), BA.4 (144 [4.1%]) and BA.5 (647 [18.3%]). Results were similar for severe adjudicated COVID-19 cases.

**Primary series vaccination.** Overall, in both groups combined, quarterly COVID-19 incidence (95% confidence intervals [CIs]) per 1000 person-months detected 14 days after injection 2 of the primary series were low, starting January 1, 2021 (0.9 [0.66–1.2]) in Part B of the trial then increased from July 1, 2021 to September 30, 2021 (8.35 [7.67–9.08]) coinciding with the Delta surge[17], with the highest incidences observed starting after December 31, 2021 (61.72 [52.89–71.60]) during the Omicron waves (Table S15). The majority of COVID-19 cases for the primary series occurred when Omicron BA.1 was predominant compared with Delta and Omicron BA.2, BA.4/5, and BQ1.1 variants (Table 1). Overall incidence starting 14 days following injection two of the primary series through April 7, 2023, were similar in the mRNA-1273 and placebo-mRNA-1273 groups for COVID-19 cases defined per the Centers for Disease Control and Prevention (CDC) guidelines[32] and prior SARS-CoV-2 infection regardless of symptomatology and severity (Table S16).

**Booster vaccination.** Among 16,368 boosted participants, COVID-19 disease incidence (95% CI per 1000 person-months) starting at 14 days post-booster was lower in the mRNA-1273 group than the placebo-mRNA-1273 group (25.43 [24.25–26.65] and 29.83 [28.46–31.24]), respectively, a finding that may be potentially related to the difference in boosting intervals between these groups (Table 2). Severe COVID-19 disease incidence was similar in the two groups (1.10 [0.87–1.36] and 1.21 [0.96–1.50], respectively) (Table 2 and S17). Most COVID-19 and severe COVID-19 incidences following the booster were detected during the Omicron BA.1, BA.2, and BA.4/5 variant waves and were highest during the BA.4/BA.5 wave. Among the 167 severe COVID-19 cases, defined per protocol and FDA guidance[1,2,33], only 1 participant was hospitalized due to COVID-19. Incidences of COVID-19 starting 14 days post-booster in the mRNA-1273 and placebo-mRNA-1273 groups, respectively, were higher in those initially pre-booster SARS-CoV-2-negative (25.65 [24.48–26.86] and 29.50 [28.17–30.88]) than those SARS-CoV-2-positive (17.02 [10.09–26.90] and 12.05 [6.41–20.60]), as were those for severe COVID-19 (1.11 [0.89–1.37] and 1.23 [0.99–1.52] vs 0.86 [0.02–4.81] and 0.00 [NE–3.22], respectively) (Table 3 and S18). Incidences of COVID-19 (Fig. 2) were generally consistent among subgroups and were higher for younger (≥18–<65 years) than older (≥65 years) participants, Hispanic/Latino versus non-Hispanic/Latino participants, and female versus male participants. Incidences of severe COVID-19 were also generally similar across subgroups, except were higher for female versus male participants and White participants versus those in communities of color (Fig. S3).

Booster receipt versus no booster was associated with relative reductions (95% CI) of 76.3% (65.7–84.1%) and 47.0% (39.0–53.9%) of COVID-19 incidences during the Delta and Omicron BA.1 waves, respectively (Table 4). The low number of non-boosted participants at risk limited evaluation in subsequent Omicron waves. These results should be interpreted with caution, given the groups were not randomized; therefore, direct comparisons may be confounded.

### Immunogenicity

Following the primary series, neutralizing antibody (nAb) GMCs (GMCs, AU/mL) against SARS-CoV-2 (D614G) decreased by day 209 post-vaccination but remained detectable over a median ~13 months follow-up prior to booster dose day 1 (BD-1) (Fig. S4). The booster dose increased nAb GMCs at BD-29 with geometric mean fold-rises (GMFRs) of 842- and 59-fold from pre-vaccination and pre-booster levels, respectively, in the PPIS-negative participants (Table 5, S19–S20 and Fig. S5). The seroresponse rates (SRRs) at BD-29 versus day 57 after the primary series were >98% (Table 5). Both the GMR and SRR differences exceeded the prespecified superiority criteria for immunogenicity of the booster compared with the primary series. These findings were irrespective of prior SARS-CoV-2 infection. Among participants with evidence of prior SARS-CoV-2 infection in the PPIS-set, GMFRs were 1907 from pre-vaccination levels and 3.2 from pre-booster levels (Tables S19–S20). Post-booster GMCs decreased by day 181 but remained higher than pre-booster levels for ≥6 months regardless of pre-booster SARS-CoV-2-infection status and age (Fig. S6). Increases in nAb titers were generally comparable regardless of sex and age, although post-booster increases in GMC from pre-booster baseline levels at day 29 were higher in SARS-CoV-2-negative participants ≥65 than those 18–65 years of age (Tables S21–S22). Results for spike binding (bAb) were similar to those of nAb (Tables S23–S24 and Fig. S7).

An analysis of pre-booster PPIS-negative participants in the mRNA-1273 group showed that nAb levels and GMFR (BD-1 to BD-29) remained generally consistent regardless of time interval (12–16 months) between second injections of mRNA-1273 and booster doses (Fig. S8). In SARS-CoV-2-positive participants at pre-booster, GMCs increased and GMFR decreased with longer time intervals; however, these changes were not statistically significant. An analysis in a subset (n = 80) of the placebo-mRNA-1273 group showed generally similar results (Fig. S9 and Table S25).

### Exploratory analyses

The effectiveness of a booster dose on COVID-19 risk was further explored in a Cox model analysis (adjusted for time-varying covariates and confounders) of ~20,000 participants in the PP-primary set who remained on study and had no COVID-19 cases by the first booster date (September 23, 2021) of the trial (Fig. S10)[34]. The proportions of

**Table 1 | Analysis of COVID-19 and severe COVID-19 incidence based on adjudication committee assessments starting 14 days after injection 2 of mRNA-1273 100-μg primary series prior to the booster, by variant waves (PP-primary series set)**

| | COVID-19 cases | | Severe COVID-19 cases | |
|---|---|---|---|---|
| | mRNA-1273 100 μg Primary series *N* = 14,291 | Placebo-mRNA-1273 100 μg Primary series *N* = 10,623 | mRNA-1273 100 μg Primary series *N* = 14,291 | Placebo-mRNA-1273 100 μg Primary series *N* = 10,623 |
| September 1, 2021–November 30, 2021 (Delta predominant) | | | | |
| Number at risk (N1) | 11,436 | 10,020 | 1,1751 | 10,179 |
| Participants with event, *n* (%)[a] | 159 (1.4) | 113 (1.1) | 27 (0.2) | 15 (0.1) |
| Person-months[b] | 20,488.2 | 19,620.5 | 21,388.1 | 20,112.0 |
| Incidence/1000 person-months (95% CI)[c] | 7.76 (6.60–9.07) | 5.76 (4.75–6.92) | 1.26 (0.83–1.84) | 0.75 (0.42–1.23) |
| December 1, 2021–March 31, 2022 (Omicron BA.1 predominant) | | | | |
| Number at risk (N1) | 1648 | 1610 | 1816 | 1714 |
| Participants with event, *n* (%)[a] | 127 (7.7) | 107 (6.6) | 8 (0.4) | 5 (0.3) |
| Person-months[b] | 2817.0 | 2230.5 | 3343.3 | 2614.6 |
| Incidence/1000 person-months (95% CI)[c] | 45.08 (37.58–53.64) | 47.97 (39.31–57.97) | 2.39 (1.03–4.72) | 1.91 (0.62–4.46) |
| April 1, 2022–June 30, 2022 (Omicron BA.2 predominant) | | | | |
| Number at risk (N1) | 399 | 271 | 506 | 356 |
| Participants with event, *n* (%)[a] | 14 (3.5) | 5 (1.8) | 1 (0.2) | 0 |
| Person-months[b] | 1120.3 | 719.8 | 1437.4 | 976.2 |
| Incidence/1000 person-months (95% CI)[c] | 12.50 (6.83–20.97) | 6.95 (2.26–16.21) | 0.70 (0.02–3.88) | 0.00 (NE–3.78) |
| July 1, 2022–November 30, 2022 (Omicron BA.4/5 predominant) | | | | |
| Number at risk (N1) | 349 | 220 | 459 | 308 |
| Participants with event, *n* (%)[a] | 8 (2.3) | 12 (5.5) | 0 | 0 |
| Person-months[b] | 1081.6 | 672.1 | 1407.2 | 940.1 |
| Incidence/1000 person-months (95% CI)[c] | 7.40 (3.19–14.57) | 17.85 (9.23–31.19) | 0.00 (NE–2.62) | 0.00 (NE–3.92) |
| December 1, 2022–April 7, 2023 (Omicron BQ.1.1 predominant) | | | | |
| Number at risk (N1) | 38 | 31 | 41 | 34 |
| Participants with event, *n* (%)[a] | 0 | 0 | 0 | 0 |
| Person-months[b] | 28.6 | 16.8 | 31.1 | 18.2 |
| Incidence/1000 person-months (95% CI)[c] | 0.00 (NE–128.91) | 0.00 (NE–220.16) | 0.00 (NE–118.69) | 0.00 (NE–203.04) |

*CI* confidence interval, *NE* not estimable.

Incidence based on COVID-19 and severe COVID-19 cases adjudicated by an independent adjudication committee to determine if the criteria for the effectiveness endpoints were met. COVID-19 is based on adjudicated cases as assessed in the primary approach for COVID-19 throughout the study. Per-protocol primary series set includes participants in Part A per-protocol set who received two doses of mRNA-1273 primary series in either Part A or Part B per schedule without major protocol deviations.

[a]Percentages are based on N1.

[b]Person-months for each time period is defined as the total months from the start of each time period or 14 days after the date of injection 2 of the mRNA-1273 primary series, whichever is later, to the date of event, one day before the date of booster if received, the end of each time period, last date of study participation, or effectiveness data-cutoff date, whichever is the earliest. 1 month = 30.4375 days.

[c]Incidence for each time period is defined as the number of participants with an event during the time period divided by the number of participants at risk during the time period and adjusted by person-months (total time at risk) in each treatment group. The 95% CI is calculated using the exact method (Poisson distribution) and adjusted by person-months.

participants who received a booster was 60% for those who received a primary series vaccination by October 31, 2021, 88% by November 30, 2021, and 97% by December 31, 2021 (Fig. S11). Baseline characteristics of the analysis population grouped according to booster timing were generally balanced and similar to those of the overall study population, although those who unblinded early also boosted early (Table S26). Boosting reduced COVID-19 risk similarly in the mRNA-1273 and placebo-mRNA-1273 groups during the Delta and Omicron waves, with hazard ratios (95% CI) of 1.24 (0.91–1.70) and 0.91 (0.69–1.22), respectively, for the pre-boost and 0.86 (0.44–1.67) and 0.76 (0.68–0.84), respectively, for the post-boost periods (Table S27). This corresponded to a non-significant risk reduction of 14% against Delta and a significant risk reduction of 24% against Omicron for the mRNA-1273 versus the placebo-mRNA-1273 groups during longer dosing intervals between the second injection and the booster for mRNA-1273 (median, 13 months) than placebo-mRNA-1273 (median, 8 months). In both groups, the reduction (95% CI) of Delta COVID-19 risk was 83% (60%–93%) post-booster, which persisted over 60 days at 83% (67–91%), regardless of time between the second injection of the

primary series and the booster dose (Fig. S12). The risk of Omicron BA.1 COVID-19 was reduced by 56% (44%–65%) immediately post-booster, with subsequent declines to 38% (28%–47%) at 60 days and 4% (−27% −28%) by 120 days. In participants ≥65 years, the effectiveness of a booster against Omicron BA.1 decreased from 86% (69%–93%) to 28% (−47%–65%) by 120 days, and in those <65 years, decreased from 50% (36%–61%) to 6% (−29%–31%) by 120 days (Fig. S13).

## Discussion

Follow-up of the COVE trial through its study-end of April 2023 showed that primary vaccination with two injections of 100-μg mRNA-1273 induced immune responses against SARS-CoV-2 that persisted through 13 months. Subsequent boosting with a 50-μg dose elicited an immune response superior to that of primary vaccination, which remained higher than after primary vaccination for ≥6 months. Primary vaccination provided protection against COVID-19 and severe COVID-19 that was durable through the Delta wave and decreased over time during the Omicron BA.1 wave. Booster vaccination was associated with a significantly lower disease incidence during both the Delta and

**Table 2 | Analysis of COVID-19 and severe COVID-19 incidence based on adjudication committee assessments starting 14 days after booster, by variant waves (Part C PP set)**

| | Part C Booster mRNA-1273 50 µg | | | |
| --- | --- | --- | --- | --- |
| | COVID-19 cases | | Severe COVID-19 cases | |
| | mRNA-1273 100 µg Primary series N = 8500 | Placebo-mRNA-1273 100 µg Primary series N = 7868 | mRNA-1273 100 µg Primary series N = 8500 | Placebo-mRNA-1273 100 µg Primary series N = 7868 |
| **≥14 days after booster** | | | | |
| Number at risk (N1) | 8238 | 7733 | 8259 | 7747 |
| Participants with event, n (%)[a] | 1747 (21.2) | 1791 (23.2) | 84 (1.0) | 83 (1.1) |
| Person-months[b] | 68,702.2 | 60,045.3 | 76,745.4 | 68,753.6 |
| Incidence/1000 person-months (95% CI)[c] | 25.43 (24.25–26.65) | 29.83 (28.46–31.24) | 1.10 (0.87–1.36) | 1.21 (0.96–1.50) |
| **September 1, 2021–November 30, 2021 (Delta predominant)** | | | | |
| Number at risk (N1) | 7023 | 6260 | 7027 | 6265 |
| Participants with event, n (%)[a] | 5 (<0.1) | 5 (<0.1) | 1 (<0.1) | 0 |
| Person-months[b] | 6543.8 | 5634.6 | 6547.4 | 5641.7 |
| Incidence/1000 person-months (95% CI)[c] | 0.76 (0.25–1.78) | 0.89 (0.29–2.07) | 0.15 (0.00–0.85) | 0.00 (NE–0.65) |
| **December 1, 2021–March 31, 2022 (Omicron BA.1 predominant)** | | | | |
| Number at risk (N1) | 8212 | 7711 | 8237 | 7730 |
| Participants with event, n (%)[a] | 672 (8.2) | 801 (10.4) | 36 (0.4) | 39 (0.5) |
| Person-months[b] | 29932.4 | 27061.7 | 31468.9 | 28930.5 |
| Incidence/1000 person-months (95% CI)[c] | 22.45 (20.79–24.21) | 29.60 (27.58–31.72) | 1.14 (0.80–1.58) | 1.35 (0.96–1.84) |
| **April 1, 2022–June 30, 2022 (Omicron BA.2 predominant)** | | | | |
| Number at risk (N1) | 7264 | 6339 | 7922 | 7116 |
| Participants with event, n (%)[a] | 579 (8.0) | 558 (8.8) | 25 (0.3) | 26 (0.4) |
| Person-months[b] | 18,401.8 | 15,895.4 | 20,836.6 | 18,615.0 |
| Incidence/1000 person-months (95% CI)[c] | 31.46 (28.95–34.14) | 35.11 (32.25–38.14) | 1.20 (0.78–1.77) | 1.40 (0.91–2.05) |
| **July 1, 2022–November 30, 2022 (Omicron BA.4/5 predominant)** | | | | |
| Number at risk (N1) | 5233 | 4411 | 6353 | 5604 |
| Participants with event, n (%)[a] | 491 (9.4) | 427 (9.7) | 22 (0.3) | 18 (0.3) |
| Person-months[b] | 13,758.7 | 11,413.5 | 17,808.1 | 15,509.5 |
| Incidence/1000 person-months (95% CI)[c] | 35.69 (32.60–38.99) | 37.41 (33.95–41.13) | 1.24 (0.77–1.87) | 1.16 (0.69–1.83) |
| **After December 1, 2022 (Omicron BQ.1.1 predominant)** | | | | |
| Number at risk (N1) | 129 | 88 | 171 | 121 |
| Participants with event, n (%)[a] | 0 | 0 | 0 | 0 |
| Person-months[b] | 65.5 | 40.2 | 84.5 | 56.9 |
| Incidence/1000 person-months (95% CI)[c] | 0.00 (NE–56.34) | 0.00 (NE–91.66) | 0.00 (NE–43.67) | 0.00 (NE–64.79) |

*CI* confidence interval, *COVID-19* coronavirus disease 2019, *NE* not estimable, *PP* per-protocol.

Incidence based on COVID-19 and severe COVID-19 cases adjudicated by an independent adjudication committee to determine if the criteria for the effectiveness endpoints were met. COVID-19 based on adjudicated assessments are as in the primary approach for COVID-19 throughout the study. Part C per-protocol set consisted of all participants who were randomized in Part A, received mRNA-1273 50-µg booster in part C after primary series per schedule (within 21–42 days after injection 1), were pre-booster SARS-CoV-2-negative and had no major protocol deviations that impact critical or key data. Part C PP set used for analysis of effect of boosting on COVID-19 risk.

[a]Percentages are based on N1.

[b]Person-months for each time period is defined as the total months from the start of each time period or the date of booster, whichever is later, to the date of event, the end of each time period, last date of study participation, or effectiveness data-cutoff date, whichever is the earliest. 1 month = 30.4375 days.

[c]Incidence for each time period is defined as the number of participants with an event during the time period divided by the number of participants at risk during the time period and adjusted by person-months (total time at risk) in each treatment group. The 95% CI is calculated using the exact method (Poisson distribution) and adjusted by person-months.

Omicron BA.1 waves, but the effectiveness against Omicron decreased over time.

The long-term safety of the primary series and booster was consistent with data reported previously for the blinded part of the trial and with subsequently reported post-authorization data[1,2]. The mRNA-1273 vaccine demonstrated an acceptable safety profile, and no new safety concerns were identified among participants who received a two-injection 100-µg primary series during the period >6 months post-vaccination and the 50-µg booster during >11 months follow-up.

Boosting in the study with the original 50-µg dose of mRNA-1273 was associated with significantly lower incidences of COVID-19 and severe COVID-19 in the Delta and Omicron BA.1 waves; however, incidences increased during the later Omicron BA.2 and BA.4/5 waves.

Adjustment for time-varying effects in an exploratory model showed that boosting extended a reduced risk of COVID-19 by 80% through 60 days post-vaccination during the Delta wave. The risk reduction against Omicron was >50% initially, but then decreased to 4% by 120 days post-boost. The decline in effectiveness seen over time was likely due to increased immune escape of Omicron variants, given that antibody levels remained substantially enhanced after boosting. These findings suggest the need for updated boosters that are closely matched to the circulating strains to increase effectiveness against emerging variants. Overall, the incidence of clinically severe disease regardless of strain was extremely low, consistent with the benefit of vaccination and booster immunization against severe COVID-19 disease observed in real-world studies[7–11]. Taken together, these data

**Table 3 | Analysis of COVID-19 incidence based on adjudication committee assessments starting 14 days after booster, by pre-booster SARS-CoV-2 status (Part C safety set)**

| | Part C booster mRNA-1273 50 µg | | | |
|---|---|---|---|---|
| | mRNA-1273 100 µg Primary series[a] Negative N = 8705 | Placebo-mRNA-1273 100 µg Primary series Negative N = 8305 | mRNA-1273 100 µg Primary series[a] Positive N = 941 | Placebo-mRNA-1273 100 µg Primary series Positive N = 1647 |
| **≥14 days after booster** | | | | |
| Number at risk (N1) | 8431 | 8031 | 120 | 119 |
| Participants with event, n (%)[a] | 1802 (21.4) | 1845 (23.0) | 18 (15.0) | 13 (10.9) |
| Person-months[b] | 70251.6 | 62537.2 | 1057.6 | 1079.1 |
| Incidence/1000 person-months (95% CI)[c] | 25.65 (24.48–26.86) | 29.50 (28.17–30.88) | 17.02 (10.09–26.90) | 12.05 (6.41–20.60) |
| **September 1, 2021–November 30, 2021 (Delta predominant)** | | | | |
| Number at risk (N1) | 7175 | 6497 | 92 | 98 |
| Participants with event, n (%)[a] | 6 (<0.1) | 5 (<0.1) | 0 | 0 |
| Person-months[b] | 6688.5 | 5840.7 | 78.8 | 78.7 |
| Incidence/1000 person-months (95% CI)[c] | 0.90 (0.33–1.95) | 0.86 (0.28–2.00) | 0.00 (NE–46.80) | 0.00 (NE–46.88) |
| **December 1, 2021–March 31, 2022 (Omicron BA.1 predominant)** | | | | |
| Number at risk (N1) | 8404 | 8005 | 115 | 119 |
| Participants with event, n (%)[a] | 699 (8.3) | 823 (10.3) | 9 (7.8) | 5 (4.2) |
| Person-months[b] | 30589.4 | 28103.2 | 399.6 | 424.0 |
| Incidence/1000 person-months (95% CI)[c] | 22.85 (21.19–24.61) | 29.29 (27.32–31.36) | 22.52 (10.30–42.76) | 11.79 (3.83–27.52) |
| **April 1, 2022–June 30, 2022 (Omicron BA.2 predominant)** | | | | |
| Number at risk (N1) | 7425 | 6595 | 106 | 106 |
| Participants with event, n (%)[a] | 590 (7.9) | 577 (8.7) | 3 (2.8) | 3 (2.8) |
| Person-months[b] | 18829.2 | 16582.4 | 302.8 | 293.9 |
| Incidence/1000 person-months (95% CI)[c] | 31.33 (28.86–33.97) | 34.80 (32.01–37.76) | 9.91 (2.04–28.96) | 10.21 (2.11–29.83) |
| **July 1, 2022–November 30, 2022 (Omicron BA.4/5 predominant)** | | | | |
| Number at risk (N1) | 5357 | 4617 | 95 | 92 |
| Participants with event, n (%)[a] | 507 (9.5) | 440 (9.5) | 6 (6.3) | 5 (5.4) |
| Person-months[b] | 14077.1 | 11968.3 | 273.9 | 280.8 |
| Incidence/1000 person-months (95% CI)[c] | 36.02 (32.95–39.29) | 36.76 (33.41–40.37) | 21.90 (8.04–47.67) | 17.80 (5.78–41.55) |
| **After December 1, 2022 (Omicron BQ.1.1 predominant)** | | | | |
| Number at risk (N1) | 135 | 95 | 3 | 2 |
| Participants with event, n (%)[a] | 0 | 0 | 0 | 0 |
| Person-months[b] | 67.5 | 42.6 | 2.5 | 1.6 |
| Incidence/1000 person-months (95% CI)[c] | 0.00 (NE–54.66) | 0.00 (NE–86.57) | 0.00 (NE–1477.37) | 0.00 (NE–2245.61) |

CI, confidence interval; COVID-19, coronavirus disease 2019; NE, not estimable.

[a]One participant from the mRNA-1273 group had a missing pre-booster SARS-CoV-2-status.

[b]Percentages are based on N1.

[c]Person-months for each time period is defined as the total months from the start of each time period to the date of event, the end of each time period, last date of study participation, or effectiveness data-cutoff date, whichever is the earliest. 1 month = 30.4375 days.

[d]Incidence for each time period is defined as the number of participants with an event during the time period divided by the number of participants at risk during the time period and adjusted by person-months (total time at risk) in each treatment group. The 95% CI is calculated using the exact method (Poisson distribution) and adjusted by person-months.

demonstrate the durable reduction of COVID-19 risk associated with primary series vaccination during early variant waves and the benefit of booster immunization, regardless of prior history of SARS-CoV-2 exposure.

The 50-µg mRNA-1273 booster dose induced nAb and bAb immune responses that were greater than those achieved by primary vaccination, regardless of prior SARS-CoV-2 infection. These higher nAb levels are consistent with demonstrated neutralization of variants following immunization with the 50-µg mRNA-1273[19–22] and modified monovalent and bivalent boosters updated with closely matching circulating variants[23–27]. Higher nAb following a booster dose has been shown to be related to a greater risk reduction of Omicron BA.1 COVID-19 in recent correlates of risk or protection analyses of the COVE trial[35–39].

The exploratory model indicated a significant reduction of Omicron-associated COVID-19 (24%), with longer dosing intervals between primary vaccination and boosting for mRNA-1273 (13 months) versus placebo-mRNA-1273 (8 months)[34]; however, a lower, non-significant reduction (14%) was observed for Delta-associated COVID-19. Prior studies have also shown that longer-term intervals between

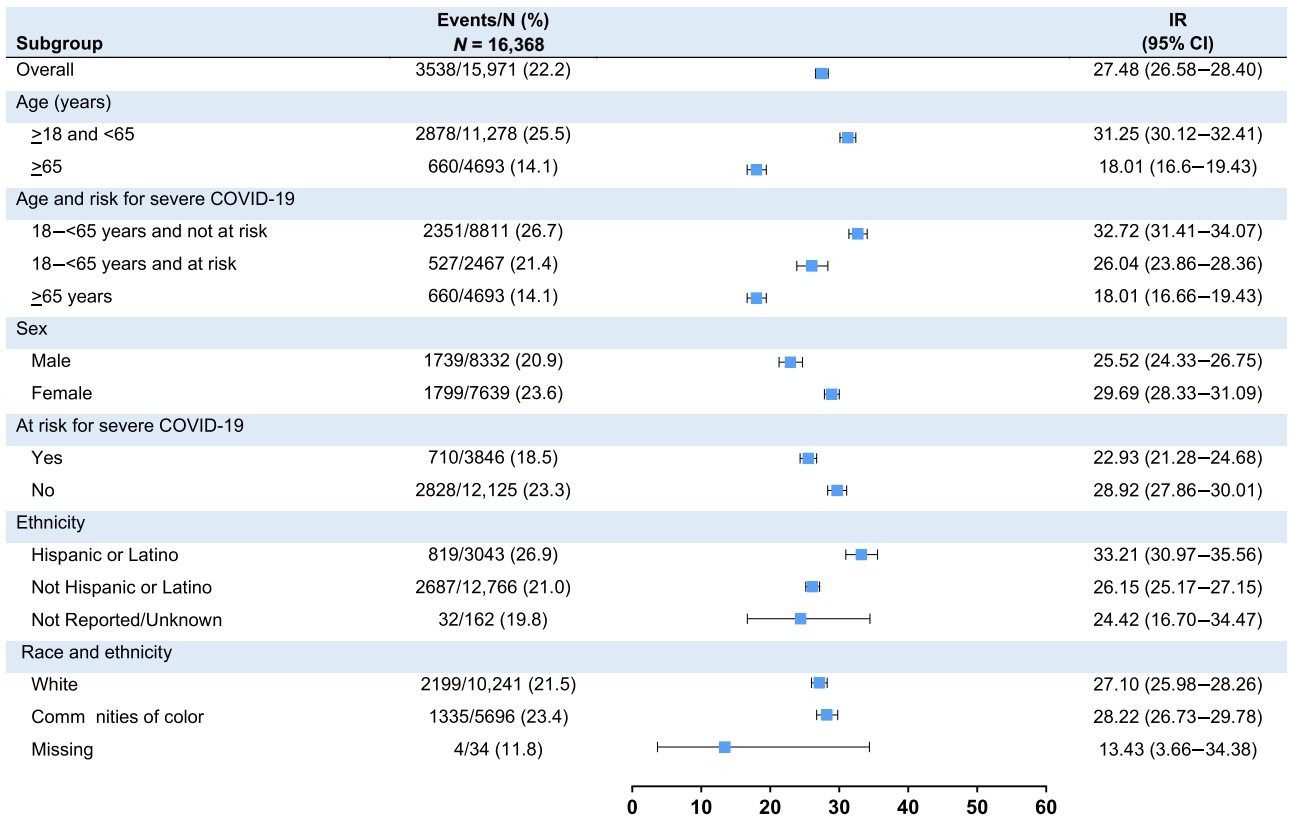

| Subgroup | Events/N (%)<br>N = 16,368 | | IR<br>(95% CI) |
|---|---|---|---|
| Overall | 3538/15,971 (22.2) | | 27.48 (26.58−28.40) |
| **Age (years)** | | | |
| ≥18 and <65 | 2878/11,278 (25.5) | | 31.25 (30.12−32.41) |
| ≥65 | 660/4693 (14.1) | | 18.01 (16.6−19.43) |
| **Age and risk for severe COVID-19** | | | |
| 18−<65 years and not at risk | 2351/8811 (26.7) | | 32.72 (31.41−34.07) |
| 18−<65 years and at risk | 527/2467 (21.4) | | 26.04 (23.86−28.36) |
| ≥65 years | 660/4693 (14.1) | | 18.01 (16.66−19.43) |
| **Sex** | | | |
| Male | 1739/8332 (20.9) | | 25.52 (24.33−26.75) |
| Female | 1799/7639 (23.6) | | 29.69 (28.33−31.09) |
| **At risk for severe COVID-19** | | | |
| Yes | 710/3846 (18.5) | | 22.93 (21.28−24.68) |
| No | 2828/12,125 (23.3) | | 28.92 (27.86−30.01) |
| **Ethnicity** | | | |
| Hispanic or Latino | 819/3043 (26.9) | | 33.21 (30.97−35.56) |
| Not Hispanic or Latino | 2687/12,766 (21.0) | | 26.15 (25.17−27.15) |
| Not Reported/Unknown | 32/162 (19.8) | | 24.42 (16.70−34.47) |
| **Race and ethnicity** | | | |
| White | 2199/10,241 (21.5) | | 27.10 (25.98−28.26) |
| Comm nities of color | 1335/5696 (23.4) | | 28.22 (26.73−29.78) |
| Missing | 4/34 (11.8) | | 13.43 (3.66−34.38) |

Incidence Rate (x-axis: 0 10 20 30 40 50 60)

**Fig. 2 | COVID-19 events based on adjudication committee assessments starting 14 days after booster, across subgroups (Part C per-protocol set).** CI confidence interval, COVID-19 coronavirus disease 2019, IR incidence rate, N number of participants at risk. Incidence rates among subgroups following administration of mRNA-1273 as a 50-µg booster dose are presented. Error bars represent 95% confidence intervals. Incidence is defined as the number of participants with an event starting 14 days after booster by the number of participants at risk at 14 days after booster and adjusted by person-months (total time at risk) in each treatment group. 1 month = 30.4375 days. White is defined as White and non-Hispanic, and communities of color include all the others whose race or ethnicity is unknown, unreported, or missing. Participants at risk for severe COVID-19 included those aged ≥65 years and <65 years with at least one of the risk factors (chronic lung disease, significant cardiac disease, severe obesity, diabetes, liver disease, or HIV infection).

vaccination and boosting result in enhanced immune responses and/or vaccine efficacy for COVID-19 and other viral diseases, including an analysis of the COVE trial[34,40–43].

The effectiveness of boosting against COVID-19 in the model analysis, as well as nAb increases from baseline, were greater among SARS-CoV-2-negative participants aged ≥65 years than those 18–65 years, consistent with age-group effects shown previously[1,44]. It is worth noting that higher risk is associated with extended time intervals prior to boosting due to the waning of primary vaccination-conferred immunity[6,44–47], and that higher numbers of older than younger individuals are likely to be boosted and may be associated with a greater risk of COVID-19 exposure[44–47]. These findings may be useful in setting policy for boost intervals in at-risk groups and are in line with risk-based COVID-19 approaches for prioritized vaccination of older adults and other at-risk groups[48,49].

The COVE study was conducted amidst the challenging times of the COVID-19 pandemic in a setting of rapidly shifting epidemiology and corresponding study changes, such as the unblinding and booster phases; nonetheless, this study was key in informing regulatory agencies toward implementation of vaccination and booster immunization strategies. A strength of this analysis over previous, observational studies is that it was based on a randomized licensure trial. Study limitations include the unblinding and resulting lack of a control group for the Part C booster phase, and the de-randomization that occurred during the study due to the availability of safe and effective vaccines through EUA, which limited

direct comparisons of disease incidence across the initial mRNA-1273 and placebo-mRNA-1273 groups. Comparisons of disease incidence are potentially impacted by the variation in background SARS-CoV-2 variant predominance for these groups, changes in individual and collective risk behavior, and in-home testing practices, as well as differences in relative timing between primary vaccination and booster immunization. Additionally, it is possible that some individuals classified as unboosted may have been essentially withdrawn from the study and thus had not reported a COVID-19 infection. Comparison between booster and non-booster participants was limited in later time periods by the fewer number of participants and person-months at risk in the non-boosted group. Interpretation of the effects of vaccination and booster immunization by SARS-CoV-2-infection status was limited by the smaller size of the SARS-CoV-2-positive compared to the SARS-CoV-2-negative group. Therefore, caution should be exercised regarding any causal interpretation of results from Parts B and C of the COVE study, as the groups were not randomized, limiting direct comparisons between study groups.

The long-term follow-up of the COVE study adds evidence that primary vaccination and boosting with mRNA-1273 provided immunogenicity and effectiveness in protection against both COVID-19 and severe COVID-19 with an acceptable safety profile, including during emergent variant waves through April 7, 2023, regardless of prior SARS-CoV-2 infection. These results are consistent with the substantial benefits achieved through global vaccination and booster immunization with safe and effective COVID-19 vaccines in controlling the

**Table 4 | COVID-19 incidence based on adjudication committee assessments by variant waves for booster and non-booster participants (PP-primary series set)**

| | COVID-19 | | Severe COVID-19 | |
|---|---|---|---|---|
| | Booster mRNA-1273 50 µg $N = 17,657$[a] | Non-Booster $N = 7257$[a] | Booster mRNA-1273 50 µg $N = 17,657$[a] | Non-Booster $N = 7257$[a] |
| **September 1, 2021–November 30, 2021 (Delta predominant)** | | | | |
| Participants at risk (N1) | 15,557 | 21,456 | 15,557 | 21,930 |
| Participants with event, n (%)[b] | 32 (0.2) | 272 (1.3) | 2 (<0.1) | 42 (0.2) |
| Person-months[c] | 19,874.5 | 40,108.7 | 19,900.3 | 41,500.1 |
| Incidence/1000 person-months[d] | 1.61 (1.10–2.27) | 6.78 (6.00–7.64) | 0.10 (0.01–0.36) | 1.01 (0.73–1.37) |
| Incidence reduction, % (95% CI)[e] | 76.3 (65.7–84.1) | | 90.1 (61.8–98.8) | |
| **December 1, 2021–March 31, 2022 (Omicron BA.1 predominant)** | | | | |
| Participants at risk (N1) | 17,549 | 3258 | 17,579 | 3530 |
| Participants with event, n (%)[b] | 1553 (8.8) | 234 (7.2) | 79 (0.4) | 13 (0.4) |
| Person-months[c] | 63,229.6 | 5047.5 | 66,803.2 | 5957.9 |
| Incidence/1000 person-months[d] | 24.56 (23.36–25.81) | 46.36 (40.61–52.70) | 1.18 (0.94–1.47) | 2.18 (1.16–3.73) |
| Incidence reduction, % (95% CI)[e] | 47.0 (39.0–53.9) | | 45.8 (−6.3–70.1) | |
| **April 1, 2022–June 30, 2022 (Omicron BA.2 predominant)** | | | | |
| Participants at risk (N1) | 15,044 | 670 | 16,541 | 862 |
| Participants with event, n (%)[b] | 1181 (7.9) | 19 (2.8) | 52 (0.3) | 1 (0.1) |
| Person-months[c] | 38,329.6 | 1840.1 | 43,691.6 | 2413.6 |
| Incidence/1000 person-months[d] | 30.81 (29.08–32.62) | 10.33 (6.22–16.12) | 1.19 (0.89–1.56) | 0.41 (0.01–2.31) |
| Incidence reduction, % (95% CI)[e] | −198.4 (−397.4 to −90.2) | | −187.3 (−11,461.5–50.7) | |
| **July 1, 2022–November 30, 2022 (Omicron BA.4/5 predominant)** | | | | |
| Participants at risk (N1) | 10,884 | 569 | 13,294 | 767 |
| Participants with event, n (%)[b] | 969 (8.9) | 20 (3.5) | 41 (0.3) | 0 |
| Person-months[c] | 28,660.7 | 1753.7 | 37,188.3 | 2347.3 |
| Incidence/1000 person-months[d] | 33.81 (31.71–36.01) | 11.41 (6.97–17.61) | 1.10 (0.79–1.50) | 0.00 (NE–1.57) |
| Incidence reduction, % (95% CI)[e] | −196.5 (−387.5 to −90.9) | | NE (NE-NE) | |
| **After December 1, 2022 (Omicron BQ.1.1 predominant)** | | | | |
| Participants at risk (N1) | 268 | 69 | 348 | 75 |
| Participants with event, n (%)[b] | 1 (0.4) | 0 | 0 | 0 |
| Person-months[c] | 127.9 | 45.4 | 165.2 | 49.2 |
| Incidence/1000 person-months[d] | 7.82 (0.20–43.57) | 0.00 (NE–81.30) | 0.00 (NE–22.34) | 0.00 (NE–74.90) |
| Incidence reduction, % (95% CI)[e] | NE (NE–NE) | | NE (NE–NE) | |

*CI* confidence interval, *COVID-19* coronavirus disease 2019, *NE* not estimable, *PP* per-protocol.

For N1 in each time period, a participant who received a booster was included in the non-booster column for the period(s) before booster, and in the booster column for the period(s) after booster. For participants with event(s) before booster and event(s) after booster, the first event after the first injection of the mRNA-1273 100-µg primary series and before booster is counted in the non-booster column; and the first event after booster is counted in the booster column. Participants who received a booster after the primary series were compared with those who had not received a booster after the primary series during the time period.

[a]N in the header presents total number of participants receiving a booster or not receiving a booster as of data-cutoff date. Participants who received a booster before the end of study participation or data-cutoff date are included in the booster group; otherwise, participants are included in the non-booster column.

[b]Percentages based on N1. For N1, participants who received a booster are included in the non-booster column for the period(s) pre- and post-booster. For participants with event(s) pre- and post-booster, the first event after the first injection of the mRNA-1273 100-µg primary series and before booster is counted in the non-booster column; and the first event after booster is counted in the booster column.

[c]For the booster column, person-months is defined as the total months from the start of each time period or the date of booster, whichever is later, to the date of the first event after booster, the end of each time period, last date of study participation, or effectiveness data-cutoff date, whichever is the earliest. For the non-booster column, person-months for each time period is defined as the total months from the start of each time period to the date of the first event after the first injection of mRNA-1273 100-µg primary series, the end of each time period, last date of study participation, or effectiveness data-cutoff date, (or one-day pre-booster date if received), whichever is the earliest.

[d]Incidence for each time period is defined as the number of participants with an event during the time period divided by the number of participants at risk during the time period and adjusted by person-months (total time at risk) in each treatment group. 95% CI is calculated using the exact method (Poisson distribution) and adjusted by person-months.

[e]Reduction in incidence rate is defined as 1−ratio of incidence rate (booster vs non-booster) and 95% CI of the ratio was calculated using the exact method conditional upon the total number of cases, adjusting for person-months for the time period.

pandemic and restoring lifestyles to pre-pandemic times[7–10]. Continual monitoring of variants and updates of COVID-19 vaccines are essential for limiting the spread of future SARS-CoV-2 infections and ensuring pandemic preparedness.

## Methods

The trial was conducted in accordance with the International Council for Harmonisation of Technical Requirements for Registration of Pharmaceuticals for Human Use, Good Clinical Practice guidelines. The

central Institutional Review Board (Advarra, Inc., Columbia, MD) approved the protocol and consent forms. All participants provided written informed consent.

### Study design and oversight

This 3-part, phase 3, observer-blind, randomized, placebo-controlled (Part A) and open-label (Parts B and C) trial was conducted in 99 US sites in adults with no or stable medical conditions as previously described with 2 years of planned follow-up (Fig. S14)[1,2]. Demographic

**Table 5 | Pseudovirus neutralizing antibody against SARS-CoV-2 (D614G) in booster Part C (BD-29 vs Part A day 57) per-protocol immunogenicity subset**

| | PPIS-negative (N = 682) | | PPIS-all participants (N = 731) | |
|---|---|---|---|---|
| | mRNA-1273 100 μg Primary series BD-29 | mRNA-1273 100 μg Primary series Day 57 | mRNA-1273 100 μg Primary series BD-29 | mRNA-1273 100 μg Primary series Day 57 |
| **BD-29** | | | | |
| n | 638 | 680 | 678 | 728 |
| GMC (95% CI)[a] | 7739.7 (7240.9–8272.8) | 1111.3 (1041.68–1185.51) | 8130.9 (7611.7–8685.6) | 1111.0 (1043.8–1182.5) |
| GMFR (vs pre-vaccination GMC[b]) (95% CI) | 841.7 (779.0–909.4) | 119.4 (110.7–128.7) | 882.4 (817.4–952.6) | 119. 1 (110.7–128.1) |
| GMFR (vs pre-booster GMC[b]) (95% CI) | 59.1 (54.6–64.1) | NA | 50.2 (45.6–55.1) | NA |
| GMR (BD-29 vs day 57 GMC) (95% CI)[c] | 7.0 (6.51–7.52) | | 7.37 (6.87–7.91) | |
| SRR from pre-vaccination (n/N1, %) (95% CI)[d] | 636/636, 100.0 (99.4–100.0) | 672/680, 98.8 (97.7–99.5) | 675/675, 100.0 (99.5–100.0) | 720/728, 98.9 (97.8–99.5) |
| SRR difference (BD day 29 vs day 57) (95% CI)[e] | 0.9 (0.1–1.8) | | 0.9 (0.1–1.7) | |
| **BD-181** | | | | |
| n | 575 | | 610 | |
| GMC (95% CI) | 2872.1 (2625.2–3142.2) | - | 2951.1 (2701.7–3223.6) | - |
| GMFR (vs pre-vaccination GMC[b]) (95% CI) | 305.9 (277.3–337.4) | - | 314.3 (285.4–346.1) | - |
| GMFR (95% CI) (vs pre-booster GMC[b]) | 21.3 (19.2–23.6) | - | 17.9 (16.0–20.2) | - |
| **Day 209** | | | | |
| n | - | 641 | - | 684 |
| GMC (95% CI) | - | 237.4 (221.9–254.0) | - | 238.2 (222.6–255.0) |
| GMFR (95% CI) | - | 25.8 (23.8–27.9) | - | 25.7 (23.8–27.9) |

*BD* booster dose *CI* confidence interval, *GMC* geometric mean concentration [AU/mL], *GMFR* geometric mean fold rise (post-baseline/baseline), *GMR* geometric mean ratio, *LLOQ* lower limit of quantification, *NA* not applicable, *RT-PCR* reverse transcriptase polymerase chain reaction (test), *SRR* seroresponse rate, *ULOQ* upper limit of quantification.

Immunogenicity was assessed by pseudovirus neutralizing antibody assay against SARS-CoV-2 (D614G) in the Part C per-protocol immunogenicity set (PPIS). The primary analysis comparison of immune responses after the booster and primary series of mRNA-1273 was performed in the PPIS-negative set, which consists of participants in the PPIS who are pre-booster SARS-CoV-2 negative (no virologic [RT-PCR at BD-D1] or serologic evidence [anti-SARS-CoV-2 nucleocapsid on/before BD-D1, Roche Elecsys] of SARS-CoV-2 infection).

*N* number of participants with non-missing data at the corresponding timepoint.

Antibody values reported as below the lower limit of quantification (LLOQ = 10) are replaced by 0.5 × LLOQ. Values greater than the upper limit of quantification (ULOQ = 111,433) are replaced by the ULOQ if actual values are not available.

[a]95% CI is calculated based on the t-distribution of the log-transformed values for GMC and GMFR, then back transformed to the original scale for presentation. 95% CI is calculated using the Clopper-Pearson method.

[b]GMCs (95% CIs) are 9.33 (8.95–9.73) at pre-vaccination (day 1) and 164.74 (148.69–182.52) at pre-booster (booster day 1).

[c]The log-transformed antibody levels are analyzed using a paired t-test method with the group variable and 95% CI is calculated based on the t-distribution of the mean of paired difference in the log-transformed values, then back transformed to the original scale for presentation.

[d]Seroresponse at a participant level is defined as a change from below the LLOQ to equal or above 4 x LLOQ, or a≥ 4-fold rise if baseline (pre-injection 1) is equal to or above the LLOQ. 95% CI is calculated using the Clopper-Pearson method.

[e]Difference in seroresponse rate and 95% CI is calculated using adjusted Wald method for the paired binary data. The number of participants included in the comparison could be different from N1.

information relating to the participant's age, sex (self-reported), race, and ethnicity was recorded at a screening in the eCRF. Parts B and C, open-label, offered participants an opportunity to receive the primary series (if they received placebo in Part A) and booster of mRNA-1273, respectively. This report is through the protocol planned follow-up, plus study augmentation with booster vaccination, given the changing response to the pandemic over time. Longer-term safety, effectiveness, and immunogenicity data from study initiation (July 27, 2020) through the open-label and booster Parts of the study (B and C) (April 7, 2023) are presented.

**Study objectives**

The efficacy, safety, and immunogenicity outcomes of the COVE trial were previously reported for the blinded Part A of the study[1,2,15]. Part B of the study provides longer-term safety follow-up and effectiveness data following the primary series from unblinding (or participant decision visit [PDV]) to BD-1. Part C objectives evaluated the safety, effectiveness, and immunogenicity following a 50-μg booster dose of mRNA-1273. Part C boosting started in Fall 2021 for all study participants when the booster immunization received EUA, thus creating a difference in boosting interval based on initial randomization schema between initial vaccine recipients (mRNA-1273 group received mRNA-

1273 July-December 20) and Part B placebo cross-over participants (placebo-mRNA-1273 group received the mRNA-1273 primary series December 2020-April 2021).

Safety for the Part A blinded portion of the study was previously reported[1,2]. Safety data for Parts B and C included unsolicited AEs for 28 days after vaccination, and medically attended (MAAEs), serious (SAEs), and AEs leading to discontinuations. In addition, safety data for Part C included AEs of special interest, including cardiac events (myocarditis and pericarditis) and vascular events for Part C through study-end. In Part C, adverse reactions (ARs) were not solicited because the reactogenicity of the 50-μg mRNA-1273 booster dose has been found to be similar to that of the second injection of the primary series[20]; ARs that met the criteria for an MAAE, SAE, or leading to discontinuation were recorded.

Effectiveness endpoints for the mRNA-1273 primary series and booster were assessed using active surveillance and included COVID-19 (COVE[1,2] and CDC[32] definitions), severe COVID-19 (as defined in the COVE protocol[1,2] and per FDA guidance[33] [Supplementary Methods]), serologically confirmed SARS-CoV-2 infection or COVID-19 regardless of symptomatology or severity, asymptomatic SARS-CoV-2 infection, and death caused by COVID-19 (Supplementary Methods). Cases of COVID-19 and severe COVID-19 were adjudicated by an independent

committee. Because Part C lacked a placebo comparison group, the effectiveness of boosting on COVID-19 risk was evaluated by a comparison of COVID-19 incidence rates in boosted and unboosted groups, as well as by inferring effectiveness based on a bridging analysis of immune responses post-boost and post-primary series.

Part C immunogenicity objectives compared immune responses following the mRNA-1273 booster at day 29 (BD-29) with immunological responses following two injections of the mRNA-1273 primary series at day 57 to infer booster effectiveness. nAb against SARS-CoV-2 were analyzed using a pseudovirus nAb assay for ancestral SARS-CoV-2 (D614G) and SARS-CoV-2 spike-bAbs using a Mesoscale Discovery (MSD) assay (Supplementary Methods)[50,51].

Exploratory endpoints included whole-genome sequence analysis of SARS-CoV-2 variants from nasopharyngeal samples of participants at illness visits and the effect of booster on COVID-19 risk using a statistical Cox model adjusting for time-varying effects that estimated COVID-19 risk reduction (Supplementary Methods).

## Statistical analysis

Statistical analysis methods are detailed in the Supplemental Methods and analysis sets and periods in Fig. S1 and Tables S28–S30. Safety was assessed for the primary series in the primary series safety set, comprised of all participants who were randomized in Part A and received at least one injection of the primary series (mRNA-1273 and placebo-mRNA-1273 groups), and for the booster in the Part C safety set consisting of randomized participants who received 50-µg mRNA-1273 booster. Summary statistics of unsolicited AEs, SAEs, MAAEs, AEs leading to withdrawal from study participation, and deaths are presented.

The longer-term effectiveness of the primary series was assessed in the per-protocol primary series (PP-primary) set, consisting of all participants who received the primary vaccination in Parts A or B and had no evidence of prior SARS-CoV-2 infection (negative RT-PCR and nucleocapsid antibody tests) prior to the primary series. COVID-19 and severe COVID-19 incidence prior to boosting in primary series recipients (mRNA-1273 [early] vs placebo-mRNA-1273 [late] vaccination) are presented. Incidence rates of endpoints (e.g, COVID-19, severe COVID-19, etc.) post-booster were assessed in the Part C-PP set, which included all participants who received the primary series in Parts A or B followed by a booster dose and were pre-booster SARS-CoV-2-infection negative with no major protocol deviations. mRNA-1273 and placebo-mRNA-1273 groups were not combined for effectiveness analyses due to differences in time intervals between the primary series and booster dose.

For the primary series and booster, incidence rates based on the number of participants with adjudicated COVID-19 and severe COVID-19 per number at risk-adjusted by person-months and 95% CI (Poisson distribution) are provided by calendar periods corresponding to variant waves (September 1, 2021–November 30, 2021 [Delta]; December 1, 2021–March 31, 2023 [Omicron BA.1]; April 1, 2022–June 30, 2022 [Omicron BA.2]; July 1, 2022–November 30, 2022 [Omicron BA.4/5]; December 1, 2022–April 7, 2023 [Omicron BQ.1.1]), and by time periods for intervals after vaccination (Supplementary Methods).

The effectiveness of the booster on COVID-19 risk was assessed in participants who received a booster after the primary series versus non-booster participants who did not receive a booster post-primary series based on adjudicated COVID-19 cases starting 14 days after the booster dose in the PP-primary set using a dynamic approach for each treatment group. Participants considered at risk in the non-boosted group were those in the PP-primary set who had not received a booster at the specified timepoint, and participants considered at risk in the booster group were those in the PP-primary set who had received a booster prior to the specified timepoint. There is an approximately 5-month difference in dosing interval between primary series and booster in the mRNA-1273 and placebo-mRNA-1273 treatment groups; thus, the model was adjusted for dosing interval using the binary factor

early prime [Yes/No], corresponding to the mRNA-1273 and placebo-mRNA-1273 treatment groups, respectively. The comparison of incidence between the booster and non-booster participants is limited, as these groups were not randomized; thus, background incidence rates could potentially be inconsistent across these groups, and direct comparisons may be confounded. Reductions in incidence (1-incidence ratio) and 95% CI were calculated using the exact method conditional upon the total number of cases, adjusting for person-months. The effectiveness of the booster against COVID-19 was also evaluated by subgroups (e.g., age, randomization risk stratification, sex, race, ethnicity, severe COVID-19 risk factor).

Immunogenicity was assessed in the Part C PPIS (Figs. S1, S2). The Part C PPIS consists of a subset of participants randomly selected in the per-protocol random subcohort for immunogenicity who received mRNA-1273 in Part A[15], were SARS-CoV-2negative at baseline (pre injection 1) and received a booster in Part C. Geometric mean concentrations (AU/mL) for nAb and GM-levels for bAb, GMFRs, and SRRs with 95% CIs (Clopper-Pearson) against ancestral SARS-CoV-2 (D614G) are provided at each post-baseline timepoint relative to pre injection 1 of the primary series and pre-booster (BD-1) baseline. Primary series and booster SRRs were defined as titer changes from baseline below the lower limit of quantification (LLOQ) to ≥4 × LLOQ, or ≥4-fold rise if the baseline is ≥LLOQ. For the primary immunogenicity objective, GM ratios (GMR) of nAb responses and SRR differences were compared at 28 days post-booster (BD-29) versus the primary series at day 57 (28 days post-injection 2) in Part C PPIS-negative participants (having no virologic/serologic evidence of SARS-CoV-2 infection on/before BD-1) and in all PPIS participants regardless of prior SARS-CoV-2-infection status. Due to the limited size of the PPIS-positive group, immunogenicity was not compared and is summarized. Criteria for non-inferiority were met when the lower bounds of the 95% CIs for the GMRs and SRR differences were >0.67 and >−10%, respectively. Criteria for superiority were met when the lower bounds of the 95% CIs for the GMRs and SRR differences were >1 and >0, respectively (Table S31). Immunogenicity data at days 209 after the primary series and 181 post-booster in the PPIS participants and by SARS-CoV-2-infection pre-booster status, as well as by age group and SARS-CoV-2-infection pre-booster status, are also presented. Analyses of the impact of time intervals between the primary series and booster on antibody responses among PPIS-negative mRNA-1273 participants and in a subset of placebo-mRNA-1273 PPIS-negative participants with non-missing time intervals and immunogenicity results at BD-1 were also performed.

Genotypic analysis of variants by whole-genome sequencing was performed on SARS-CoV-2-positive nasopharyngeal swabs collected at illness visits from participants following the primary series and the booster in Part C. Samples were collected from July 2020 (start of Part A) through April 2022 (data-cutoff date date). The number and percentage of cases by variant were summarized.

The effect of booster on COVID-19 risk was further explored using a Cox proportional hazards model in participants in the PP-primary set who remained on study and were COVID-19-naive as of September 23, 2021 (first date that participants were boosted); those who did not receive a booster through January 31, 2022, were censored on that date (see details in the Supplementary Methods and Figs. S10, S11)[34]. The model adjusted for baseline factors (e.g., sex, stratification factor for severe COVID-19 risk [≥18 years and <65 years old not at risk, ≥18 years and <65 years old at risk, and ≥65 years old], risk score, early unblinding). As boosting was not randomized, the mRNA-1273 and placebo-mRNA-1273 groups were evaluated in terms of boost initiation assessed using early and late boosted participants who received booster before and on or later than October 25, 2021 (median of booster dates) as a covariate of time since vaccination in the model. The incidences of symptomatic COVID-19 against Delta and Omicron during the follow-up period from September 23, 2021, through April 5, 2022, were calculated

starting at 14 days post-booster; those who acquired COVID-19 during the interval from the booster dose up to 13 days later were censored. Evaluation of the effectiveness of booster vaccination in the Delta wave period was limited to 60 days from the date of the first booster (September 23, 2021) during the time of Delta variant circulation prior to the start of the Omicron variant circulation period (approximately early December 2021).

All analyses were conducted using SAS Version 9.4 or higher.

## Reporting summary

Further information on research design is available in the Nature Portfolio Reporting Summary linked to this article.

## Data availability

Data associated with this study are provided in the manuscript and/or Supplementary Appendix; the protocol and statistical analysis plan are provided as Supplementary Information. Individual-level data reported in this study involving human research participants are not publicly shared due to potentially identifying or sensitive patient information. Access to participant-level data and supporting clinical documents by qualified external researchers may be made available upon request and subject to review. A materials transfer and/or data access agreement with the sponsor will be required for accessing shared data. Such requests can be made to Dr. Lindsey Baden, Brigham and Women's Hospital, Boston, MA 02115, USA.

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

## Acknowledgements

We thank the dedicated participants of the COVE trial, the members of the trial team, members of the DSMB (Richard J. Whitley [Chair], University of Alabama School of Medicine; Abdel Babiker, MRC Clinical Trials Unit at University College, London; Lisa Angeline Cooper, Johns Hopkins University School of Medicine and Bloomberg School of Public Health; Susan Smith Ellenberg, University of Pennsylvania; Alan Fix, Vaccine Development Global Program Center for Vaccine Innovation and Access PATH; Marie Griffin, Vanderbilt University Medical Center; Steven Joffe, Perelman School of Medicine, University of Pennsylvania; Jorge Kalil, Clinics Hospital [HC-FMUSP], Universidade de São Paulo, Brazil; Myron M. Levine, University of Maryland School of Medicine; Malegapuru William Makgoba, University of KwaZulu-Natal; Anastasios A. Tsiatis, North Carolina State University; Renee H. Moore, Emory University) and the adjudication committee (Richard J. Hamill [Chair], Baylor College of Medicine; Lewis Lipsitz, Harvard Medical School; Eric S Rosenberg, Massachusetts General Hospital: Anthony Faugno, Tufts Medical Center) for their critical and timely review of the trial data. We also acknowledge the contribution from the mRNA-1273 Product Coordination Team from BARDA (Robert Bruno, Richard Gorman, Holli Hamilton, Gary Horwith, Chuong Huynh, Nutan Mytle, Corrina Pavetto, Xiaomi Tong, and John Treanor), and Frank J. Dutko (Moderna consultant) for figure development and editorial support. The trial sponsor, Moderna, Inc. was responsible for conceptualization and overall trial design (in collaboration with the Biomedical Advanced Research and Development Authority [BARDA], NIAID, Coronavirus Vaccine Prevention Network, and study co-chairs), site selection and monitoring, data analysis, preparation of the manuscript, and decision to publish. This work was supported in whole or in part with federal funds from the Department of Health and Human Services; Administration for Strategic Preparedness and Response; Biomedical Advanced Research and Development Authority, under Contract No. 75A50120C00034 (Moderna, Inc.). The findings and conclusions herein are those of the authors and do not necessarily represent the views of the Department of Health and Human Services or its components. The NIAID provides grant funding to the HIV Vaccine Trials Network (HVTN) Leadership and Operations Center (UM1 AI68614HVTN), the Statistics and Data Management Center (UM1 AI68635), the HVTN Laboratory Center. (UM1 AI68618), the HIV Prevention Trials Network Leadership and Operations Center (UM1 AI68619, UM1 AI068614), the AIDS Clinical Trials Group Leadership and Operations Center (UM1 AI68636), and the Infectious Diseases Clinical Research Consortium Leadership Group 5 (UM1 AI148684-03).

## Author contributions

J.M.M., H.Z., L.R.B., B.E., H.M.E.S., D.F., K.N., B.G., and L.C. contributed to the design of the study and were responsible for clinical trial data acquisition in collaboration with G.H., C.S., J.S.O., S.D.L., J.A.W., E.J.A., and L.C. and the other COVE study investigators (Supplementary information). L.R.B., H.M.E.S., B.E., D.F., K.N., F.P., J.E.T., M.B., B.G., D.S., V.U., X.W., W.D., H.Z., A.D., R.D., and J.M.M. contributed to the analysis and/or interpretation of data in relation to the manuscript. L.R.B., H.M.E.S., D.F., F.P., J.E.T., X.W., and A.D. developed the initial draft. The authors vouch for the accuracy and completeness of the data and the fidelity of the trial to the protocol. All authors contributed to the review and editing of the manuscript and approved the final version for submission.

## Competing interests

The authors report the following competing interests. L.R.B. and H.M.E.S. report grants from NIH and/or NIAID for the conduct of this study; E.J.A. reports grants from GSK Janssen, MedImmune, Merck,

Micron Technology, Pfizer, and Sanofi Pasteur, served as a consultant for Janssen, Medscape, Pfizer and Sanofi Pasteur and as a member of DSMBs for Kentucky Bioprocessing and Sanofi Pasteur; K.N. reports receiving a grant from Pfizer; J.S.O. and L.C. report receiving NIH grants; E.J.A., F.P., M.B., B.G., D.S., V.U., X.W., W.D., H.Z., A.D., R.D., and J.M.M. report being employees of Moderna, Inc., and may hold stock/stock options in the company and J.E.T. is a Moderna, Inc. consultant. Authors B.E., D.F., G.H., C.S., J.A.W., and S.D.L. declare no competing interests.

## Additional information

**Lindsey R. Baden** [1,13] ✉, **Hana M. El Sahly** [2,13], **Brandon Essink**[3], **Dean Follmann** [4], **Gregory Hachigian**[5], **Cynthia Strout**[6], **J. Scott Overcash**[7], **Susanne Doblecki-Lewis** [8], **Jennifer A. Whitaker**[2], **Evan J. Anderson** [9], **Kathleen Neuzil** [10], **Lawrence Corey** [11], **Frances Priddy**[12], **Joanne E. Tomassini** [12], **Mollie Brown**[12], **Bethany Girard**[12], **Dina Stolman**[12], **Veronica Urdaneta**[12], **Xiaowei Wang**[12], **Weiping Deng**[12], **Honghong Zhou**[12], **Avika Dixit** [12], **Rituparna Das**[12], **Jacqueline M. Miller**[12] & the COVE Trial Consortium

[1]Brigham and Women's Hospital, Boston, MA, USA. [2]Baylor College of Medicine, Houston, TX, USA. [3]Meridian Clinical Research Omaha, Omaha, NE, USA. [4]National Institute of Allergy and Infectious Disease, Bethesda, MD, USA. [5]Benchmark Research, Sacramento, CA, USA. [6]Coastal Carolina Research Center, Mount Pleasant, SC, USA. [7]Velocity Clinical Research, San Diego, CA, USA. [8]University of Miami, Miami, FL, USA. [9]Emory University School of Medicine, Atlanta, GA, USA. [10]University of Maryland, Baltimore, MD, USA. [11]Fred Hutchinson Cancer Research Center, Seattle, WA, USA. [12]Moderna, Inc., Cambridge, MA, USA. [13]These authors contributed equally: Lindsey R. Baden, Hana M. El Sahly. ✉e-mail: lbaden@bwh.harvard.edu

## the COVE Trial Consortium

**Lindsey R. Baden** [1,13], **Hana M. El Sahly** [2,13], **Brandon Essink**[3], **Dean Follmann** [4], **Gregory Hachigian**[5], **Cynthia Strout**[6], **J. Scott Overcash**[7], **Susanne Doblecki-Lewis** [8], **Jennifer A. Whitaker**[2], **Evan J. Anderson** [9] ✉, **Kathleen Neuzil** [10] & **Lawrence Corey** [11]

A full list of members and their affiliations appears in the Supplementary Information.

