## [Peer Review File · Nature Communications]

REVIEWER COMMENTS

Reviewer #1 (Remarks to the Author):

Review Nature communications

Baden et al. report the results of the open-label parts, including administration of active primary vaccination to placebo participants after unblinding (part B) and administration of a booster dose (part C) of the COVE trial, which was the pivotal phase III trial (part A, randomized and blinded) for the EUA of the mRNA-1273 vaccine in December 2020.

Efficacy and safety results of part A have been previously published in the NEJM; the long-term results of Part B as well as the results of part C in the present manuscript are novel. They add additional evidence to the results that have been accumulated from other studies (including "real world" data after regulatory approvals) since 2020.

The manuscript is well written, with detailed numerical results presented in the supplementary appendix. The authors succeed well in communicating the combination of designs (part B and C) in the manuscript. This study has the strengths of a prospective trial design with standardized procedures and endpoints definition, and a large sample size.

As stated by the authors in some sections of the manuscript and appendix, parts B and C are no longer a randomized design (in contrast to part A), thus making all comparisons potentially prone to bias. The authors provide a clear rationale and context for these design choices (and there was indeed no evident alternative choice at the time of conduct of parts B and C). The authors show appropriate detailed descriptions and stratified analyses with regards to measured confounders, and also an exploratory Cox model adjusting for time-varying confounders. However, as in other uncontrolled designs, residual (unmeasured) confounding cannot be excluded, in particular confounding related to changes in individual risk-mitigating behaviours (social distancing, mask wearing etc) over time. However, the concordance with results from other studies is a strength for the overall evidence generated since 2020 (throughout studies). The publication of the present results from the COVE trial are thus important.

Major comment :

Related to potential confounding in these uncontrolled parts of the study, I would recommend that the authors add an additional word of caution with regards to causality in Parts B and C of the COVE trial to the discussion section of the manuscript, and critically re-assess the wording used throughout the manuscript ; the wording in some paragraphs is a rather confirmative, causal wording, which seems to strong.

Specific comments in line with my general comment above :

- Lines 150-153 : Mention here that these comparisons may be confounded (this is mentioned in the Table legend but is also worth mentioning in the text).

Is it sensible to show the incidence reduction estimate in Table 4 as this pushes towards a comparative interpretation ?

- Line 175 : According to Figure S9, this is not statistically significant.

- Line 189: the 14% reduction for Delta is not significant according to the confidence interval of the relative efficacy ratio in table S32

- Lines 190-191 : isn't the model adjusted for the dosing interval between primary series and boost (see

lines 588-589)? Is the covariate a continuous variable or binary (not completely clear from lines 588-589)

- Lines 192 and 241 (and maybe elsewhere in the manuscript) : I suggest not to use the word « efficacy » here (given the uncontrolled design and potential confounding).
- Line 207 and line 215 : « was significantly associated with lower incidence » would be a more cautious wording (instead of « significantly reduced incidence »)
- Lines 256-259 : and also changes in (individual and collective) risk behaviour ? Furthermore, an additional sentence of precaution with regards to a causal interpretation of the Part B and C results could be added in this paragraph.
- Lines 264 : « adds evidence » (rather than « shows ») ?
- Lines 549-559 : see my thoughts above whether any incidence reduction estimates should be shown, as these drive towards a comparative interpretation

Minor comments :

Some inconsistencies or errors seem to be present in the text and tables, and need a thorough review by the authors to correct potential errors.

- Table S19 : the denominators for the percentages shown in this table do not seem consistent, i.e. BA.1 91.7% “severe cases” ; denominator is inconsistent –(should be 102, and thus 10.8% to maintain consistency across the table)
- lines 147-148 : isn’t the incidence higher in females than in males ? (see Fig S4)

Additional minor comments to improve clarity for the reader :

- If feasible, add some numerical results to the abstract
- Line 59 : add date of study end
- Line 157 : explain « BD-1 » at its first occurrence in the text (it is currently only explained in the legends and supplemental appendix)
- Line 179 : Briefly mention adjustment for time-varying confounders also here in the text (not only in the methods and supplement)
- Figure 1 : During the transition from Part A to Part B in the mRNA active arm (as randomized in part A), it seems that quite a large number were unblinded between the 1st and the 2nd injection of the primary series (and thus received the 2nd injection open-label in part B). Is this correct ? This could be stated more clearly in the Figure and in its legend. The notion of “two second injections” (mentioned in the legend) is not clear to me.
- Suppl appendix : If possible, add a table of contents to the supplemental appendix, with hyperlinks to facilitate navigation in this document
- Figure S2 : Why is the safety analysis set in panel B different from the “received at least one injection” analysis set? Is this due to vaccine administration errors (where true received injection differed from randomized group) ? It would be helpful to add this more clearly in the flow chart (or its legend)
- Figure S9 and S10: These could be easier to follow if both figures had the same format. One analyzes fold rise at D29 post-boost whereas the other analyses absolute titers at D29
- Table S5 : specify more clearly what « baseline » refers to in the title of the table. It seems to be the Part A baseline but some information in this table concern the booster baseline or the PDV.
- Tables S24 and S25 : the difference between these two supplemental tables is not completely clear to me (Part C is also shown in Table S24; although the analysis population seems to be slightly different

from table S25).

- Table S32 : mention all covariates in the legend. The table's content is not sufficiently self-explanatory. The pre-boost placebo-mRNA seems to be the reference category and all other HRs are relative to this ?

Reviewer #2 (Remarks to the Author):

The manuscript of Lindsey R. Baden et al. 'Long-term safety and efficacy of COVE study open-label and booster phases' provides a significant portion of safety, immunogenicity, and exploratory efficacy readouts from the pivotal licensure-enabling Phase 3 study of Moderna's mRNA-1273 vaccine. The presented materials are comprehensive and detailed and provide a better understanding of vaccine performance against antigenically diverse SARS-CoV-2 variants after the primary vaccination and booster dose. Despite the absence of an appropriate control group, the authors found an elegant solution using a dynamic approach for each treatment group. Multiple subgroup safety analyses are helpful and further characterize the benefit-risk profile of the vaccine in various subpopulations.

The manuscript is well-written and would interest a broad audience of healthcare providers, policy decision-makers, physicians, and vaccine experts.

There are several questions and considerations:

The study title, 'Long-term safety and efficacy of COVE study open-label and booster phases,' might be confusing. Assuming the authors present 'Long-term safety and efficacy of mRNA-1273: results from open-label and booster study phases.'

The authors refer to vaccine efficacy in the title and across the manuscript but present the incidence of COVID-19 disease for different periods of SARS-CoV-2 circulation and incidence reduction based on exploratory analysis.

Lines 109-110: The authors mentioned that four deaths were associated with COVID-19, two during Part B and 2 after the booster in Part C. Including short information about these cases (at least MedDRA PTs) would be beneficial.

Lines 132-136: The authors described a lower incidence of COVID-19 starting from 14 days post-booster in participants from the mRNA-1273 group compared to the placebo-mRNA-1273 group. This difference is statistically significant. The rates of severe COVID-19 also appeared to be lower in the mRNA-1273 group. The authors may want to explain this observation (different interval from primary vaccination, selection bias?)

Lines 18-139: Among 167 severe COVID-19 cases, only 1 participant was hospitalized due to COVID-19. Please specify which criteria of severe COVID-19 (as per protocol/US FDA guidance) were applied for these subjects.

Lines 155-165 and Tables S5-S8 – please indicate if GMC using PPD SARS-CoV-2 (D614G) Neutralizing Antibody (VAC62) assay is presented in WHO-calibrated units (IU/mL) or arbitrary units

Lines 191-195 and Figure S13—the results of exploratory analysis represent a quick reduction of vaccine efficacy against the Omicron BA.1 variant after a booster dose compared to a no-booster approach. The Delta variant results look different and sustain 60-80% efficacy for 60 days post booster. However, no data beyond 60 days are presented. Some explanation might be helpful.

Line 218-219: The phrase 'that boosting extended a reduced risk of COVID-19 by 80% through 60 days post-vaccination during the Delta wave, and initially by >50% against Omicron, but then decreased

during the period 120 days post-boost' may require simplification. It's unclear whether a reduced risk is transient for Omicron variants only or applicable for all variants with significant antigenic distance from the vaccine strain.

Figure 1 – there are several discrepancies identified:

149 subjects disappeared from the subject disposition (1725 started the open-label study, 1576 discontinued, 25 completed study)

1727 subjects discontinued study (1728 are listed under all categories)

417 subjects disappeared (12554 received the second dose – 2185 discontinued the study – 9952 entered in Part C)

15184 subjects in mRNA-1273, 14661 started the open-label study, 9647 received a second injection (WHICH INJECTION? Per protocol, no cross-over vaccination was provided to mRNA-1273 group participants)

524 subjects missed (14661 started open-label study, 4490 discontinued, 9647 started Part C)

Tables 1 and 2—Why is the number of subjects at risk for severe COVID-19 disease higher than the number of subjects at risk for any COVID-19?

Table 2 - Were the subjects who developed COVID-19 at an early post-booster period excluded from subsequent analysis?

Table 4 – How authors can explain a negative incidence reduction for periods of dominant circulation of BA.2 and BA.4/5? The number of subjects in a non-booster cohort (569-670) is large enough and cannot wholly explain a 2-3-fold difference.

Lines 483-488 – reference to Figure S12 might be beneficial to show progression with booster immunization in both groups. A similar Figure presenting coverage with 1st and 2nd vaccination in the mRNA-1273 group and cross-over vaccination in the placebo-mRNA-1273 group would also be helpful.

Reviewer #3 (Remarks to the Author):

The paper presents the results of the booster extension of the COVE trial, a randomised controlled trial of mRNA-1273 booster dose efficacy. Given rapidly changing epidemiology, the study considered the protective effect of booster vaccines over rapid emergence of the BA.1 Omicron variant. The study reports that a lower infection incidence was seen in the group that received a longer interval between primary vaccination and boosters and demonstrates larger immune responses to ancestral strains following boosting. Most countries have now recommended up to three further vaccine boosters in vulnerable populations, based on findings from large observational studies however there have been few controlled trials particularly in the Omicron dominant period. This study adds to the existing field of evidence by providing robust high-quality data from a controlled trial with reliable reporting of adverse events as a recent.

Minor points:

Where were study participants recruited from?

Although the supplement describes the SARS-CoV-2 PCR testing that participants underwent, the schedule is not completely clear. Can the authors comment on whether all participants had the same

access to testing (symptomatic and asymptomatic)?

Twelve of the authors including the senior author are affiliated with the vaccine manufacturer, it would be worth mentioning this in the limitations section.

How were participants recruited and randomised? What were the exclusion and inclusion criteria?

Please add a description of the study design and the point estimates of effect sizes to the abstract.

Figure 1 – not all participants are accounted for. Within the placebo arm 1725 started open label – 25 completed but only 1576 shown as discontinued so there are 124 unaccounted for. Placebo crossed over arm – 12554 received 2nd injection, 9952 entered part C but only 2185 who discontinued are accounted for.

It would be useful to report the level of precision when reporting AEs in order to compare the groups.

As described in lines 73-75, prior infection prevalence at baseline seems very low in the pre-booster groups. How does this compare to the prevalence of prior infection in the general population?

Line 107 describes that 65.4% of participants experienced AEs before the study end, 23.6% were related to the vaccine – can the authors provide further detail of what these were?

It is interesting that more severe infections were seen in white compared with individuals from other ethnicities – this is contrary to much of the published literature. Can the authors comment on this?

The CIs for hazards ratios described in lines 189-191 of mRNA-1273 included 1 and were very wide suggesting low precision and lack of statistical significance. It would be important to note this in lines 236-238 of the discussion that make reference to this result.

It is interesting that booster had longer duration of effectiveness in >65 vs <65 group (lines 197-198), can the authors comment on this?

In line with the SAGER guidelines, please can the authors present the results by sex and report how sex was taken into account in the study design.

Lines 222-223 - These findings support the need for boosting to increase protection against emerging variants. As the authors hypothesise that infections occurred due to immune escape variants, it would be worth mentioning that boosters need to be updated to include most recent variants as well.

A number of long sentences that may be easier to understand if broken up (ie lines 201-205, 241-245).

Figure 2 – define IR, how was “at risk of severe COVID-19” defined?

Table 5 – variation in decimal places reported

Reviewer 1

Baden et al. report the results of the open-label parts, including administration of active primary vaccination to placebo participants after unblinding (part B) and administration of a booster dose (part C) of the COVE trial, which was the pivotal phase III trial (part A, randomized and blinded) for the EUA of the mRNA-1273 vaccine in December 2020.

Efficacy and safety results of part A have been previously published in the NEJM; the long-term results of Part B as well as the results of part C in the present manuscript are novel. They add additional evidence to the results that have been accumulated from other studies (including "real world" data after regulatory approvals) since 2020.

The manuscript is well written, with detailed numerical results presented in the supplementary appendix. The authors succeed well in communicating the combination of designs (part B and C) in the manuscript. This study has the strengths of a prospective trial design with standardized procedures and endpoints definition, and a large sample size.

As stated by the authors in some sections of the manuscript and appendix, parts B and C are no longer a randomized design (in contrast to part A), thus making all comparisons potentially prone to bias. The authors provide a clear rationale and context for these design choices (and there was indeed no evident alternative choice at the time of conduct of parts B and C). The authors show appropriate detailed descriptions and stratified analyses with regards to measured confounders, and also an exploratory Cox model adjusting for time-varying confounders. However, as in other uncontrolled designs, residual (unmeasured) confounding cannot be excluded, in particular confounding related to changes in individual risk-mitigating behaviours (social distancing, mask wearing etc) over time. However, the concordance with results from other studies is a strength for the overall evidence generated since 2020 (throughout studies). The publication of the present results from the COVE trial are thus important.

Major Comments

Comment 1. Related to potential confounding in these uncontrolled parts of the study, I would recommend that the authors add an additional word of caution with regards to causality in Parts B and C of the COVE trial to the discussion section of the manuscript, and critically re-assess the wording used throughout the manuscript ; the wording in some paragraphs is a rather confirmative, causal wording, which seems to strong.

Specific comments in line with my general comment above:

- Lines 150-153: Mention here that these comparisons may be confounded (this is mentioned in the Table legend but is also worth mentioning in the text).

Response: We thank the Reviewer for their feedback and have accordingly included the clarification text in the Methods and Results sections.

Revised text:

Methods

[Lines 629-632]: *“The comparison of incidence between booster and non-booster participants is limited, as these groups were not randomized; thus, background incidence rates could potentially be inconsistent across these groups and direct comparisons may be confounded.”*

Results

[Lines 161–165]: *“Booster receipt versus no booster was associated with relative reductions (95% CI) of 76.3% (65.7-84.1%) and 47.0% (39.0-53.9%) of COVID-19 incidences during the Delta and Omicron BA.1 waves, respectively (Table 4). The low number of non-boosted participants at risk limited evaluation in subsequent Omicron waves. These results should be interpreted with caution given the groups were not randomized, therefore direct comparisons may be confounded.”*

Comment 2. Is it sensible to show the incidence reduction estimate in Table 4 as this pushes towards a comparative interpretation ?

Response: Thank you for the comment. We have retained this information as we believe it provides valuable information on incidence rate reductions during specific SARS-CoV-2 waves. To avoid comparative interpretation, we added clarifying text on the limitations of this analysis to the methods as above in comment 1.

Revised text:

Methods

[Lines 629-632]: *“The comparison of incidence between booster and non-booster participants is limited, as these groups were not randomized; thus, background incidence rates could potentially be inconsistent across these groups and direct comparisons may be confounded.”*

Comment 3. Line 175: According to Figure S9, this is not statistically significant.

Response: Thank you for the comment. We have accordingly clarified this in the text.

Revised text:

Results

[Lines 184–188]: *“An analysis of pre-booster PPIS-negative participants in the mRNA-1273 group showed that nAb levels and GMFR (BD-1 to BD-29) remained generally consistent regardless of time interval (12-16 months) between second injections of mRNA-1273 and booster doses (Fig. S9). In SARS-CoV-2-positive participants at pre-booster, GMCs increased and GMFR decreased with longer time intervals; however, these changes were not statistically significant.”*

Comment 4. Line 189: the 14% reduction for Delta is not significant according to the confidence interval of the relative efficacy ratio in table S32

Response: Thank you for the comment. We have revised the statement to clarify this.

Revised text:

Results

[Lines 203–207]: *“This corresponded to a non-significant risk reduction of 14% against Delta and a significant risk reduction of 24% against Omicron for the mRNA-1273 versus the placebo-mRNA-1273 arm during longer dosing intervals between the second injection and the booster for mRNA-1273 (median, 13 months) than placebo-mRNA-1273 (median, 8 months).”*

Comment 5. Lines 190-191: isn't the model adjusted for the dosing interval between primary series and boost (see lines 588-589)? Is the covariate a continuous variable or binary (not completely clear from lines 588-589)

Response: There is an approximately 5-month difference in dosing interval between primary series and boost in mRNA-1273 and placebo-mRNA-1273 treatment groups, thus the model was adjusted for the dosing interval using the binary factor “early prime [Yes/No],” corresponding to the mRNA-1273 and placebo-mRNA-1273 treatment groups respectively

Revised text:

Methods

[Lines 626–629]: *“There is an approximately 5-month difference in dosing interval between the primary series and booster in the mRNA-1273 and placebo-mRNA-1273 treatment groups; thus, the model was adjusted for dosing interval using the binary factor early prime [Yes/No], corresponding to the mRNA-1273 and placebo-mRNA-1273 treatment groups, respectively.”*

Comment 6. Lines 192 and 241 (and maybe elsewhere in the manuscript): I suggest not to use the word « efficacy » here (given the uncontrolled design and potential confounding).

Response: Thank you for the comment. We have revised efficacy to effectiveness throughout the manuscript, where appropriate, including revision of the title.

Revised text:

Title

Long-term safety and effectiveness of mRNA-1273 vaccine in adults: COVE trial open-label and booster phases

Results

[Line 121]: *“Efficacy subheading” revised to “Effectiveness”*

[Lines 192–195]: *“The effectiveness of a booster dose on COVID-19 risk was further explored in a Cox model analysis (adjusted for time-varying covariates and confounders) of ~20,000 participants in the PP-primary set who remained on study and had no COVID-19 cases by the first booster date (September 23, 2021) of the trial (Fig. S11).³⁴”*

[Lines 207–212]: *“In both groups, the reduction (95% CI) of Delta COVID-19 risk was 83% (60%–93%) post-booster which persisted over 60 days at 83% (67%-91%), regardless of time*

between the second injection of the primary series and the booster dose (Fig. S13). The risk of Omicron BA.1 COVID-19 was reduced by 56% (44%–65%) immediately post-booster, with subsequent declines to 38% (28%–47%) at 60 days and 4% (-27%–28%) by 120 days. In participants ≥65 years, the effectiveness of a booster against Omicron BA.1 decreased from 86% (69%–93%) to 28% (-47%–65%) by 120 days, and in those <65 years, decreased from 50% (36%–61%) to 6% (-29%–31%) by 120 days (Fig. S14)."

Discussion

[Lines 222–224]: *"Booster vaccination was associated with a significantly lower disease incidence during both the Delta and Omicron BA.1 waves, but the effectiveness against Omicron decreased over time."*

[Lines 234–237]: *"Adjustment for time-varying effects in an exploratory model showed that boosting extended a reduced risk of COVID-19 by 80% through 60 days post-vaccination during the Delta wave. The risk reduction against Omicron was >50% initially, but then decreased to 4% by 120 days post-boost. The decline in effectiveness seen over time was likely due to increased immune escape of emerging variants, given that antibody levels remained substantially enhanced after boosting."*

[Lines 260–266]: *"The effectiveness of boosting against COVID-19 in the model analysis as well as nAb increases from baseline were greater among SARS-CoV-2-negative participants aged ≥65 years than those 18-65 years, consistent with age-group effects shown previously.^{1,44} It is worth noting that higher risk is associated with extended time intervals prior to boosting due to waning of primary vaccination-conferred immunity,⁴⁴⁻⁴⁸ and that higher numbers of older than younger individuals are likely to be boosted and may be associated with a greater risk of COVID-19 exposure.⁴¹⁻⁴⁷"*

Methods

Study design and oversight

[Lines 547–549]: *"Longer-term safety, effectiveness, and immunogenicity data from study initiation (July 27, 2020) through the open-label and booster Parts of the study (B and C) (April 7, 2023) are presented."*

Study Objectives

[Lines 556–558]: *"Part B of the study provides longer-term safety follow-up and effectiveness data following the primary series from unblinding (or participant decision visit [PDV]) to BD-1. Part C objectives evaluated the safety, effectiveness, and immunogenicity following a 50-μg booster dose of mRNA-1273."*

[Lines 574–579]: *"Effectiveness endpoints for the mRNA-1273 primary series and booster were assessed using active surveillance and included COVID-19 (COVE^{1,13} and CDC³² definitions), severe COVID-19 (as defined in the COVE protocol^{1,13} and per FDA guidance³³ [Supplementary Methods]), serologically confirmed SARS-CoV-2 infection or COVID-19 regardless of*

symptomatology or severity, asymptomatic SARS-CoV-2 infection and death caused by COVID-19 (Supplementary Methods)."

[Lines 580–583]: "Because Part C lacked a placebo comparison group, the effectiveness of boosting on COVID-19 risk was evaluated by a comparison of COVID-19 incidence rates in boosted and unboosted groups, as well as by inferring effectiveness based on a bridging analysis of immune responses post-boost and post-primary series."

Statistical analysis

[Lines 601–604]: "The longer-term effectiveness of the primary series was assessed in the per-protocol primary series (PP-primary) set, consisting of all participants who received the primary vaccination in Parts A or B and had no evidence of prior SARS-CoV-2 infection (negative RT-PCR and nucleocapsid antibody tests) prior to the primary series."

[Lines 634–636]: "The effectiveness of the booster against COVID-19 was also evaluated by subgroups (e.g., age, randomization risk stratification, sex, race, ethnicity, severe COVID-19 risk factor)."

Comment 7. Line 207 and line 215: « was significantly associated with lower incidence » would be a more cautious wording (instead of «significantly reduced incidence »)

Response: We have accordingly revised the text in the Discussion section.

Revised text:

Discussion

[Lines 222–224]: "Booster vaccination was associated with a significantly lower disease incidence during both the Delta and Omicron BA.1 waves, but the effectiveness against Omicron decreased over time."

[Lines 231–233]: "Boosting in the study with the original 50- μ g dose of mRNA-1273 was associated with significantly lower incidences of COVID-19 and severe COVID-19 in the Delta and Omicron BA.1 waves; however, incidences increased during the later Omicron BA.2 and BA.4/5 waves."

Comment 8. Lines 256-259: and also changes in (individual and collective) risk behaviour? Furthermore, an additional sentence of precaution with regards to a causal interpretation of the Part B and C results could be added in this paragraph.

Response: Thank you for the comment. We have incorporated the suggested revisions in the Discussion section.

Revised text:

Discussion

[Lines 269–289]: "The COVE study was conducted amidst the challenging times of the COVID-19 pandemic in a setting of rapidly shifting epidemiology and corresponding study changes, such as the unblinding and booster phases; nonetheless, this study was key in informing regulatory agencies toward implementation of vaccination and booster immunization strategies. A strength of this analysis over previous, observational studies is that it was based on a randomized

licensure trial. Study limitations include the unblinding and resulting lack of a control group for the Part C booster phase, the de-randomization that occurred during the study due to the availability of safe and effective vaccines through EUA, which limited direct comparisons of disease incidence across the initial mRNA-1273 and placebo-mRNA-1273 groups. Additionally, comparisons of disease incidence are potentially impacted by the variation in background SARS-CoV-2 variant predominance for these groups, changes in individual and collective risk behavior, and in home testing practices, as well as in relative timing between primary vaccination and booster immunization. Additionally, it is possible that some individuals classified as unboosted may have been essentially withdrawn from the study and thus not reported a COVID-19 infection. Comparison between booster and non-booster participants was limited in later time periods by the fewer number of participants and person-months at risk in the non-boosted group. Interpretation of the effects of vaccination and booster immunization by SARS-CoV-2-infection status was limited by the smaller size of the SARS-CoV-2-positive compared to the SARS-CoV-2-negative group. Therefore, caution should be exercised regarding any causal interpretation of results from Parts B and C of the COVE study, as the groups were not randomized, limiting direct comparisons between study arms.”

Comment 9. Lines 264: « adds evidence » (rather than « shows ») ?

Response: We have accordingly revised this text in the Discussion section.

Revised text:

Discussion

[Lines 290–293]: “The long-term follow-up of the COVE study adds evidence that primary vaccination and boosting with mRNA-1273 provided immunogenicity and effectiveness in protection against both COVID-19 and severe COVID-19 with an acceptable safety profile, including during emergent variant waves through April 7, 2023, regardless of prior SARS-CoV-2 infection.”

Comment 10. Lines 549-559: see my thoughts above whether any incidence reduction estimates should be shown, as these drive towards a comparative interpretation

Response: As mentioned above, we prefer to provide the incidence reduction estimates and appropriate limitations have been stated.

Minor Comments

Comment 11. Some inconsistencies or errors seem to be present in the text and tables, and need a thorough review by the authors to correct potential errors.

Table S19: the denominators for the percentages shown in this table do not seem consistent, i.e. BA.1 91.7% “severe cases” ; denominator is inconsistent –(should be 102, and thus 10.8% to maintain consistency across the table)

Response: Thank you, Table S19 has been revised and other tables checked throughout.

Comment 12. lines 147-148: isn't the incidence higher in females than in males ? (see Fig S4)

Response: Thank you for bringing this our attention. We have corrected the details in the text.

Revised text:

Results

[Lines 158–160]: *“Incidences of severe COVID-19 were also generally similar across subgroups, except were higher for female versus male participants and White participants versus those in communities of color (Fig. S4).”*

Additional minor comments to improve clarity for the reader:

Comment 13. If feasible, add some numerical results to the abstract

Response: We have added key data points to the abstract within the word count.

Revised text:

Abstract

[Lines 26–41]: *“Coronavirus Efficacy (COVE) was a 3-part, phase 3, observer-blind, randomized, placebo-controlled (Part A) and open-label (Parts B and C) trial. Vaccination with two injections of mRNA-1273 (100 µg) was shown to be safe and efficacious at preventing coronavirus disease 2019 (COVID-19) at completion of the blinded part of the COVE study (Part A). We present the final report of the longer-term safety and efficacy data of the primary vaccination series plus a 50-µg booster dose administered in Fall 2021 (Parts B and C). The booster safety profile was consistent with that reported for the primary series. Incidences of COVID-19 and severe COVID-19 were higher during the Omicron BA.1 than Delta variant waves, and boosting versus non-boosting was associated with a significant, 47.0% (95% CI: 39.0-53.9%) reduction of Omicron BA.1 COVID-19 incidence (24.6 [23.4–25.8] vs 46.4 [40.6–52.7]). In an exploratory Cox regression model adjusted for time-varying covariates, a longer median interval between primary vaccination and boosting for the mRNA-1273 (13 months) versus the placebo-mRNA-1273 (8 months) arm was associated with a significantly lower, 24.0% (16.0%–32.0%) risk of COVID-19 during the Omicron BA.1 wave. Boosting elicited greater immune responses against ancestral SARS-CoV-2 than the primary series, irrespective of prior SARS-CoV-2 infection.”*

Comment 14. Line 59: add date of study end

Response: We have included the details in the text.

Revised text:

Introduction

[Lines 63–65]: *“Herein, the final results of the open-label and booster Parts (B and C) of the trial through study-end (April 2023) are reported.”*

Comment 15. Line 157: explain « BD-1 » at its first occurrence in the text (it is currently only explained in the legends and supplemental appendix)

Response: We have clarified the abbreviation at its first mention in the text.

Revised text:

Results

[Lines 167–170]: *“Following the primary series, neutralizing antibody (nAb) geometric mean concentrations (GMCs, AU/mL) against SARS-CoV-2 (D614G) decreased by day 209 post-vaccination, but remained detectable over a median ~13 months follow-up prior to booster dose day 1 (BD-1) (Fig. S5).”*

Comment 16. Line 179: Briefly mention adjustment for time-varying confounders also here in the text (not only in the methods and supplement)

Response: The changes have been incorporated in the text.

Revised text:

Results

[Lines 192–195]: *“The effectiveness of a booster dose on COVID-19 risk was further explored in a Cox model analysis (adjusted for time-varying covariates and confounders) of ~20,000 participants in the PP-primary set who remained on study and had no COVID-19 cases by the first booster date (September 23, 2021) of the trial (Fig. S11).”³⁴*

Comment 17. Figure 1: During the transition from Part A to Part B in the mRNA active arm (as randomized in part A), it seems that quite a large number were unblinded between the 1st and the 2nd injection of the primary series (and thus received the 2nd injection open-label in part B). Is this correct ? This could be stated more clearly in the Figure and in its legend. The notion of “two second injections” (mentioned in the legend) is not clear to me.

Response: Among 15,185 participants randomized in the mRNA arm in Part A, 14,661 entered open label arm (ie, had Participant Decision Visits or non-missing unblinding dates); however, these participants did not receive any injection in the Part B open-label phase. We have clarified this in Figure 1 and the corresponding legend.

Comment 18. Suppl appendix: If possible, add a table of contents to the supplemental appendix, with hyperlinks to facilitate navigation in this document

Response: A table of contents has been added to the Supplementary Appendix.

Comment 19. Figure S2: Why is the safety analysis set in panel B different from the “received at least one injection” analysis set? Is this due to vaccine administration errors (where true received injection differed from randomized group)? It would be helpful to add this more clearly in the flow chart (or its legend)

Response: Yes, the difference of participant numbers between the safety set and full analysis set is due to randomization errors. The vaccination groups for the safety set correspond to the actual injections that participants received, while the treatment groups for the full analysis set

correspond to the injections to which participants were randomized. This was defined in the Figure S2 legend and has been revised for better clarity.

Revised text:

Supplementary Appendix

[Fig. S2 legend]: *"The full analysis set includes all participants who were randomized and received at least one injection of the mRNA-1273 primary series or placebo in Part A and corresponds to the randomized study vaccinations. The primary series safety set includes all participants who were randomized and received at least one injection of the mRNA-1273 primary series or placebo in Part A and corresponds to the actual study vaccinations that participants received; the primary series safety set was used for the safety analysis of the mRNA-1273 primary series."*

Comment 20. Figure S9 and S10: These could be easier to follow if both figures had the same format. One analyzes fold rise at D29 post-boost whereas the other analyses absolute titers at D29

Response: Thank you for the comment. Figure S9 displays absolute titers in panel A and the fold-rise in titers in panel B for mRNA-1273 participants only in the prespecified Part C PPIS (per protocol immunogenicity set of participants who received mRNA-1273 in Part A, were SARS-CoV-2 negative at baseline [pre-injection 1] and received a booster in Part C). Figure S9 aims to show how pre-booster status impacts the absolute titers and the fold-rise post booster. Figure S10 is provided to show immunogenicity data at BD-D1 pre-booster and BD-D29 post-booster for mRNA-1273 PPIS participants including a small subset (n=80) of placebo-mRNA-1273 who were also pre-booster negative at baseline as absolute titers. There is no intention to show fold rise as it has been demonstrated in Figure S9. The Y axes labels and graph titles have been revised for Figure S10 for better clarity.

Revised text: See revised Figure S10 Y axes and titles in the Supplement.

Comment 21. Table S5: specify more clearly what « baseline » refers to in the title of the table. It seems to be the Part A baseline but some information in this table concern the booster baseline or the PDV.

Response: The title for Table S5 and legend have been revised for clarity.

Revised text:

Supplementary Appendix

[Table S5 title]: *"Baseline Demographics Prior to Primary Vaccination, Part C Safety Set."*

[Table S5 footnote]: *"Baseline characteristics were captured at enrollment in part A, prior to primary vaccination."*

Comment 22. Tables S24 and S25: the difference between these two supplemental tables is not completely clear to me (Part C is also shown in Table S24; although the analysis population seems to be slightly different from table S25).

Response: Tables S24 and S25 provide nAb data against ancestral SARS-CoV-2 (D614G) at each post-baseline timepoint relative to pre-injection 1 of primary series (Part A) and pre-booster (BD-1) baseline, respectively. We have clarified this in the Table S24 and 25 legends.

Revised text:

Supplementary Appendix

[Table S24 footnote]: *“Pre-vaccination baseline was evaluated prior to primary vaccination injection 1 on day 1 (Part A).”*

[Table S25 footnote]: *“Pre-booster baseline was evaluated prior to booster vaccination on BD-day 1.”*

Comment 23. Table S32: mention all covariates in the legend. The table’s content is not sufficiently self-explanatory. The pre-boost placebo-mRNA seems to be the reference category and all other HRs are relative to this ?

Response: The model was adjusted for baseline factors (sex, stratification factor at randomization, risk score, early unblinding), as described in the Methods and Supplementary methods section. We have now added a description to the Table S32 legend.

Revised text:

Supplementary Appendix

[Table S32 footnote]: *“The boosting effectiveness on COVID-19 risk was assessed in Cox models as described in the Methods and Supplementary Methods, by randomized study vaccination group and boost period using the covariate (sex, stratum, risk score, early unblinding) adjustment for potential confounders.”*

Reviewer 2

The manuscript of Lindsey R. Baden et al. 'Long-term safety and efficacy of COVE study open-label and booster phases' provides a significant portion of safety, immunogenicity, and exploratory efficacy readouts from the pivotal licensure-enabling Phase 3 study of Moderna's mRNA-1273 vaccine. The presented materials are comprehensive and detailed and provide a better understanding of vaccine performance against antigenically diverse SARS-CoV-2 variants after the primary vaccination and booster dose. Despite the absence of an appropriate control group, the authors found an elegant solution using a dynamic approach for each treatment group. Multiple subgroup safety analyses are helpful and further characterize the benefit-risk profile of the vaccine in various subpopulations. The manuscript is well-written and would interest a broad audience of healthcare providers, policy decision-makers, physicians, and vaccine experts.

There are several questions and considerations:

Comment 1. The study title, 'Long-term safety and efficacy of COVE study open-label and booster phases,' might be confusing. Assuming the authors present 'Long-term safety and efficacy of mRNA-1273: results from open-label and booster study phases.'

Response: We thank the Reviewer for the comment. We have accordingly revised the title of the manuscript within the word limit. We also revised efficacy to effectiveness.

Revised text:

[Title page]: *"Long-term safety and effectiveness of mRNA-1273 vaccine in adults: COVE trial open-label and booster phases."*

Comment 2. The authors refer to vaccine efficacy in the title and across the manuscript but present the incidence of COVID-19 disease for different periods of SARS-CoV-2 circulation and incidence reduction based on exploratory analysis.

Response: We thank the Reviewer for this comment. We have revised sections of the manuscript results pertaining to Part C and booster exploratory analysis to note effectiveness instead of efficacy as these parts did not have a placebo. However, we have retained the term efficacy elsewhere as appropriate.

Revised text:

Results

[Lines 192–195]: *"The effectiveness of a booster dose on COVID-19 risk was further explored in a Cox model analysis (adjusted for time-varying covariates and confounders) of ~20,000 participants in the PP-primary set who remained on study and had no COVID-19 cases by the first booster date (September 23, 2021) of the trial (Fig. S11)."*

[Lines 207–210]: *"In both groups, the reduction (95% CI) of Delta COVID-19 risk was 83% (95% CI: 60%–93%) post-booster which persisted over 60 days at 83% (67%–91%), regardless of time between the second injection of the primary series and the booster dose. The risk of Omicron BA.1 COVID-19 was reduced by 56% (44%–65%) immediately post-booster, with subsequent*

declines to 38% (28%-47%) at 60 days and 4% (-27%–28%) by 120 days (Fig. S13). In participants ≥ 65 years, the effectiveness of a booster against Omicron BA.1 decreased from 86% (69%–93%) to 28% (-47%–65%) by 120 days, and in those < 65 years, decreased from 50% (36%–61%) to 6% (-29%–31%) by 120 days (Fig. S14)."

Discussion

[Lines 222–224]: *"Booster vaccination was associated with a significantly lower disease incidence during both the Delta and Omicron BA.1 waves, but effectiveness against Omicron decreased over time."*

[Lines 237–238]: *"The decline in effectiveness seen over time was likely due to increased immune escape of Omicron variants, given that antibody levels remained substantially enhanced after boosting."*

Comment 3. Lines 109-110: The authors mentioned that four deaths were associated with COVID-19, two during Part B and 2 after the booster in Part C. Including short information about these cases (at least MedDRA PTs) would be beneficial.

Response: Thank you for the comment. We have included the MedDRA preferred terms for these cases in the manuscript in the main text and provided additional details in Supplementary Table S12 footnote.

Revised text:

Results

[Lines 117–120]: *"Four deaths were associated with COVID-19, two during Part B (COVID-19 and pneumonia bacterial, and acute respiratory failure) and two after booster in Part C (respiratory failure [updated to COVID-19 after database lock] and COVID-19)."*

Supplementary Appendix

[Table S12 footnote]: *"§Four deaths were associated with COVID-19. Two deaths occurred in participants aged ≥ 60 years during Part B; one attributed to COVID-19 and pneumonia bacterial in a male on open-label day 293 post-vaccination, and another due to acute respiratory failure on open-label day 115 post-vaccination in a female who had an ongoing life-threatening COVID-19 infection with onset 1 month prior to the fatal event. Two deaths in male participants aged ≥ 65 years occurred after the booster in Part C, attributed to respiratory failure (updated to COVID-19 after database lock) on study day 98 after the booster dose, and COVID-19 (symptomatic COVID-19) on day 39 after the booster dose, respectively."*

Comment 4. Lines 132-136: The authors described a lower incidence of COVID-19 starting from 14 days post-booster in participants from the mRNA-1273 group compared to the placebo-mRNA-1273 group. This difference is statistically significant. The rates of severe COVID-19 also appeared to be lower in the mRNA-1273 group. The authors may want to explain this observation (different interval from primary vaccination, selection bias?)

Response: Thank you for the comment. As pointed by the Reviewer, the difference in the COVID-19 incidence rates between mRNA-1273 versus placebo-mRNA-1273 group could be explained by the different time periods of primary vaccination and interval between second

dose and booster. Part C boosting started in Fall 2021 for all study participants when the booster immunization received Emergency Use Authorization, thus creating a difference in boosting interval based on initial randomization schema between initial vaccine recipients (mRNA-1273 group received mRNA-1273 July-December 20) and Part B placebo crossover participants (placebo-mRNA-1273 group received the mRNA-1273 primary series December 2020-April 2021). We have accordingly included this as a potential explanation for the difference in COVID-19 incidence rates.

Revised text

Results

[Lines 142–145]: *“Among 16,368 boosted participants, COVID-19 disease incidence (95% CI per 1000 person-months) starting at 14 days post-booster was lower in the mRNA-1273 group than the placebo-mRNA-1273 group (25.43 [24.25-26.65] and 29.83 [28.46-31.24]), respectively, a finding that may be potentially related to the difference in boosting intervals between these groups.”*

Comment 5. Lines 18-139: Among 167 severe COVID-19 cases, only 1 participant was hospitalized due to COVID-19. Please specify which criteria of severe COVID-19 (as per protocol/US FDA guidance) were applied for these subjects.

Response: Severe COVID-19 was defined per the COVE protocol^{1, 2} and FDA guidance³ as the first occurrence of COVID-19 starting 14 days after the second dose of vaccine or placebo AND any of the following: clinical signs indicative of severe systemic illness: respiratory rate ≥ 30 per minute, heart rate ≥ 125 beats per minute, SpO₂ $\leq 93\%$ on room air at sea level or PaO₂/FIO₂ < 300 mmHg, OR respiratory failure or acute respiratory distress syndrome (defined as needing high-flow oxygen, non-invasive or mechanical ventilation, or extracorporeal membrane oxygenation), evidence of shock (systolic BP < 90 mmHg, diastolic BP < 60 mmHg or requiring vasopressors), OR significant acute renal, hepatic, or neurologic dysfunction, OR admission to an intensive care unit or death. Severe COVID-19 is defined in detail in the online protocol provided in the Supplementary Appendix, in addition to the blinded portion of the previously reported COVE efficacy trial. We have also cited these references for severe COVID-19 in the results and methods section.

Revised text:

Results

[Lines 149–150]: *“Among the 167 severe COVID-19 cases, defined per protocol and FDA guidance,^{1,13,33} only 1 participant was hospitalized due to COVID-19.”*

Methods

[Lines 574–579]: *“Effectiveness endpoints for the mRNA-1273 primary series and booster were assessed using active surveillance and included COVID-19 (COVE^{1,13} and CDC³² definitions), severe COVID-19 (defined in COVE protocol^{1,13} and per FDA guidance³² Supplementary Methods),”*

Supplementary Methods

[Lines 252–261]: *“Severe COVID-19 was defined per the COVE protocol^{2,3} and FDA guidance⁴ as the first occurrence of a confirmed case of COVID-19 as per the primary efficacy endpoint”*

starting 14 days after the second dose of vaccine or placebo AND any of the following: 1) clinical signs indicative of severe systemic illness, respiratory rate ≥ 30 per minute, heart rate ≥ 125 beats per minute, oxygen saturation $\leq 93\%$ on room air at sea level or arterial partial pressure of oxygen/fraction of inspired oxygen < 300 mmHg, OR 2) respiratory failure or acute respiratory distress syndrome (defined as needing high-flow oxygen, non-invasive or mechanical ventilation, or extracorporeal membrane oxygenation), evidence of shock (systolic blood pressure < 90 mmHg, diastolic BP < 60 mmHg or requiring vasopressors), OR 3) significant acute renal, hepatic or neurologic dysfunction, OR 4) admission to an intensive care unit or death.”

Comment 6. Lines 155-165 and Tables S5-S8 – please indicate if GMC using PPD SARS-CoV-2 (D614G) Neutralizing Antibody (VAC62) assay is presented in WHO-calibrated units (IU/mL) or arbitrary units

Response: The GMC are provided in AU/ml. This has been added to the results text at first mention, methods and the table legends.

Revised text:

Results

[Lines 167–170]: *“Following the primary series, neutralizing antibody (nAb) geometric mean concentrations (GMCs, AU/mL) against SARS-CoV-2 (D614G) decreased by day 209 post-vaccination, but remained detectable over a median ~ 13 months follow-up prior to booster dose day 1 (BD-1) (Fig. S5).”*

Methods

[Lines 640–643]: *“Geometric mean concentrations (AU/mL) for nAb and GM-levels for bAb, GMFRs, and SRRs with 95% CIs (Clopper-Pearson) against ancestral SARS-CoV-2 (D614G) are provided at each post-baseline timepoint relative to pre-injection 1 of the primary series and pre-booster (BD-1) baseline.”*

Comment 7. Lines 191-195 and Figure S13—the results of exploratory analysis represent a quick reduction of vaccine efficacy against the Omicron BA.1 variant after a booster dose compared to a no-booster approach. The Delta variant results look different and sustain 60-80% efficacy for 60 days post booster. However, no data beyond 60 days are presented. Some explanation might be helpful.

Response: The presentation of Delta data 60 days post booster was due to the fact that there was an approximately 2- to 3-month period from the date of first booster (September 23, 2021) when the Delta variant circulation was ongoing to when the Omicron variant started (approximately early December 2021). Thus, performing estimates of vaccine efficacy against Delta beyond the start point of Omicron variant circulation may have been misleading. We have added clarifying text to the methods.

Revised text:

Methods

[Lines 678–681]: *“Evaluation of the effectiveness of booster vaccination in the Delta wave period was limited to 60 days from the date of the first booster (September 23, 2021) during the time of Delta variant circulation prior to the start of the Omicron variant circulation period (approximately early December 2021).”*

Comment 8. Line 218-219: The phrase ‘that boosting extended a reduced risk of COVID-19 by 80% through 60 days post-vaccination during the Delta wave, and initially by >50% against Omicron, but then decreased during the period 120 days post-boost’ may require simplification. It’s unclear whether a reduced risk is transient for Omicron variants only or applicable for all variants with significant antigenic distance from the vaccine strain.

Response: Thank you for the comment. We have simplified this statement and clarified that a decrease in the effectiveness of the booster was observed against Omicron variants.

Revised text:

Discussion

[Lines 234–240]: *“Adjustment for time-varying effects in an exploratory model showed that boosting extended a reduced risk of COVID-19 by 80% through 60 days post-vaccination during the Delta wave. The risk reduction against Omicron was >50% initially, but then decreased to 4% by 120 days post-boost. The decline in effectiveness seen over time was likely due to increased immune escape of Omicron variants, given that antibody levels remained substantially enhanced after boosting. These findings suggest the need for updated boosters that are closely matched to the circulating strains to increase effectiveness against emerging variants.”*

Comment 9. Figure 1 – there are several discrepancies identified:

149 subjects disappeared from the subject disposition (1725 started the open-label study, 1576 discontinued, 25 completed study)

1727 subjects discontinued study (1728 are listed under all categories)

417 subjects disappeared (12554 received the second dose – 2185 discontinued the study – 9952 entered in Part C)

15184 subjects in mRNA-1273, 14661 started the open-label study, 9647 received a second injection (WHICH INJECTION? Per protocol, no cross-over vaccination was provided to mRNA-1273 group participants)

524 subjects missed (14661 started open-label study, 4490 discontinued, 9647 started Part C)

Response: We thank the Reviewer for bringing this to our attention. We have addressed these discrepancies, and updated Figure 1 and the legend, to clarify the participant disposition and accounting. Please note:

Placebo group: 1725 participants started open-label Part B; 1576 discontinued and 25 completed part B (1725–1576–25=124). Of the remaining 124 participants, 115 entered part C and did not receive a booster, 10 entered part C and received a booster (one of which did not enter the open-label observational phase), 7 discontinued part C and 3 completed part C.

Placebo-mRNA-1273 group: 12,649 participants started open-label Part B; 2185 discontinued and 116 completed part B. Of the remaining 10,348 participants who subsequently entered Part C, 9952 received the booster and 395 did not; 1727 discontinued (one individual discontinued but was not counted as such because their discontinuation occurred after the cutoff date).

12649-2185-116-395(-1)=9952. Of the 9952 who received a booster, 1727 discontinued from study and 8225 completed part C.

mRNA-1273 group: 14,661 participants started open-label Part B; 4490 discontinued and 114 participants completed part B. Of the remaining 10,057 participants who entered Part C, 9647 received a booster (included 10 participants who did not enter the open-label observational phase) and 420 did not (14661-4490-114-420=9637+10=9647); 1184 discontinued study and 8463 completed part C.

Revised Figure 1 legend:

[Lines 488–506]: “Figure 1. Trial profile of participants in the Part parts B and C Safety sets.

BDV, booster decision visit; FPFV, first participant first visit; PDV, participant decision visit.

**Participants (n=29,035) who started the open-label observational phase include those who had a PDV or unblinding date. †In Part B, 12,554 of the 12,649 participants in the placebo-mRNA-1273 group who received a first injection of mRNA-1273 also received a second dose. Of the 15,185 participants in the mRNA-1273 group who received at least one injection of mRNA-1273 in Part A, 139 received one injection of mRNA-1273 in Part B. §One participant discontinued after the discontinuation cutoff date. ¶A greater proportion of participants in the placebo compared with the mRNA-1273 group discontinued the study due to protocol deviations, primarily receipt of an off-study COVID-19 vaccine. ||The higher number of discontinuations by withdrawal of consent in the mRNA-1273 group is explained largely by recruitment of participants to other booster dose clinical studies (~3,863 participants to phase 2 [NCT04405076] and phase 2/3 [NCT05249829] studies). ‡Participants were considered to have completed the study if they completed the final visit at day 759 (month 25), 24 months following the last injection of study vaccination. **Included one participant who did not enter the open-label observational phase. ††Included 10 participants who did not enter the open-label observational phase. Study initiation dates: for Part A blinded FPFV July 27, 2020; for Part B (PDV) December 2020; Part C (BDV) September 23, 2021. Data cutoff date: Part A March 26, 2021; Part B booster-day 1 visit or database lock date, April 7, 2023, whichever is earlier; database lock date: Part C April 7, 2023.”*

Comment 10. Tables 1 and 2—Why is the number of subjects at risk for severe COVID-19 disease higher than the number of subjects at risk for any COVID-19?

Response: "At risk" is defined as participants in the specified analysis sets who had no COVID-19/severe COVID-19 infection prior to the start of the summary period. In general, there were fewer severe COVID-19 events than COVID-19 events, and consequently higher numbers of participants at risk for severe COVID-19.

Comment 11. Table 2 - Were the subjects who developed COVID-19 at an early post-booster period excluded from subsequent analysis?

Response: Participants who developed COVID-19 at an early post-booster period were excluded from subsequent analysis, as these participants were not counted as “at risk” for subsequent periods.

Comment 12. Table 4 – How authors can explain a negative incidence reduction for periods of dominant circulation of BA.2 and BA.4/5? The number of subjects in a non-booster cohort (569-670) is large enough and cannot wholly explain a 2-3-fold difference.

Response: This result is because incidence reduction compares the incidences in non-booster vs boosted participants. For BA.2 and BA.4/5, the reduction is greater in the non-boosted than boosted group. This is likely attributed to the differences in timing intervals between primary vaccination and boosting in the boosted versus non-boosted participants. At the time of BA.2 (01Apr 2022- 30Jun2022) circulation and BA.4/5 (01Jul2022-30Nov2022), the majority of participants have been boosted for 4-6 months, and the boosting effect has significantly waned. In addition, behavioral differences between boosted and non-boosted participants may have potentially affected the incidence rates. We state this limitation in the discussion and have revised per reviewer 1 comment #8 above.

Revised text:

Discussion

[Lines 277–283]: *“Comparisons of disease incidence are potentially impacted by the variation in background SARS-CoV-2 variant predominance for these groups, changes in individual and collective risk behavior and in home testing practices, as well as in relative timing between primary vaccination and booster immunization. Additionally, it is possible that some individuals classified as unboosted may have been essentially withdrawn from the study and thus not reported a COVID-19 infection.”*

Comment 13. Lines 483-488 – reference to Figure S12 might be beneficial to show progression with booster immunization in both groups. A similar Figure presenting coverage with 1st and 2nd vaccination in the mRNA-1273 group and cross-over vaccination in the placebo-mRNA-1273 group would also be helpful.

Response: Thank you for the comment. We have accordingly cited Fig. S12 in the Methods section.

Revised text:

Methods

[Lines 559–565]: *“Part C boosting started in Fall 2021 for all study participants when the booster immunization received EUA, thus creating a difference in boosting interval based on initial randomization schema between initial vaccine recipients (mRNA-1273 group received mRNA-1273 July-December 20) and Part B placebo crossover participants (placebo-mRNA-1273 group received the mRNA-1273 primary series December 2020-April 2021). The cumulative proportion of boosted participants between September 2021 and March 2022 is presented Fig. S12.”*

Reviewer 3

The paper presents the results of the booster extension of the COVE trial, a randomised controlled trial of mRNA-1273 booster dose efficacy. Given rapidly changing epidemiology, the study considered the protective effect of booster vaccines over rapid emergence of the BA.1 Omicron variant. The study reports that a lower infection incidence was seen in the group that received a longer interval between primary vaccination and boosters and demonstrates larger immune responses to ancestral strains following boosting. Most countries have now recommended up to three further vaccine boosters in vulnerable populations, based on findings from large observational studies however there have been few controlled trials particularly in the Omicron dominant period. This study adds to the existing field of evidence by providing robust high-quality data from a controlled trial with reliable reporting of adverse events as a recent.

Minor points:

Comment 1. Where were study participants recruited from?

Response: Thank you for the comment. As previously published^{1, 2}, the COVE study enrolled participants from 99 centers in the United States.

Comment 2. Although the supplement describes the SARS-CoV-2 PCR testing that participants underwent, the schedule is not completely clear. Can the authors comment on whether all participants had the same access to testing (symptomatic and asymptomatic)?

Response: Surveillance for COVID-19 was performed through weekly contact with participants via a combination of telephone calls and completion of an eDiary starting at day 1 through the end of the study. Participants with symptoms of COVID-19 lasting at least 48 hours (except for fever and/or respiratory symptoms) returned to the clinic or were visited at home by medically qualified site staff within 72 hours to collect a nasopharyngeal swab sample for RT-PCR testing for SARS-CoV-2 and other respiratory pathogens. Alternatively, if a clinic or home visit was not possible, the participants themselves submitted a saliva sample for RT-PCR testing. We have included this information in the Supplementary Methods section.

Revised text:

Supplementary Methods

[Lines 147–153]: *“Surveillance for COVID-19 was performed through weekly contact with participants via a combination of telephone calls and completion of an eDiary starting at day 1 through the end of the study. Participants with symptoms of COVID-19 lasting at least 48 hours (except for fever and/or respiratory symptoms) returned to the clinic or were visited at home by medically qualified site staff within 72 hours to collect a nasopharyngeal swab sample for RT-PCR testing for SARS-CoV-2 and other respiratory pathogens. Alternatively, if a clinic or home visit was not possible, the participants themselves submitted a saliva sample for RT-PCR testing.”*

Comment 3. Twelve of the authors including the senior author are affiliated with the vaccine manufacturer, it would be worth mentioning this in the limitations section.

Response: Thank you for the comment. We have added to the author masthead that the listed coauthors represent the large COVE Study Group (inadvertently omitted from the co-author list) and note that The Cove Study Group and Trial Investigators and Study Teams are listed in the supplement for PubMed indexing, in accordance with requirements Good Publication Practice (GPP) Guidelines for Company-Sponsored Biomedical Research: 2022 Update (<https://pubmed.ncbi.nlm.nih.gov/36037471/>). We have also added the role of the sponsor and coauthors to the contribution section of the manuscript. Competing interests of authors are reported at the end of the manuscript.

Revised text:

Author masthead

[Lines 3–8]: *Lindsey R. Baden^{1*}, Hana M. El Sahly^{2*}, Brandon Essink³, Dean Follmann⁴, Gregory Hachigian⁵, Cynthia Strout⁶, J. Scott Overcash⁷, Susanne Doblecki-Lewis⁸, Jennifer A. Whitaker², Evan J. Anderson^{9**}, Kathleen Neuzil¹⁰, Lawrence Corey¹¹, Frances Priddy¹², Joanne E Tomassini¹², Mollie Brown¹², Bethany Girard¹², Dina Stolman¹², Veronica Urdaneta¹², Xiaowei Wang¹², Weiping Deng¹², Honghong Zhou¹², Avika Dixit¹², Rituparna Das¹², Jacqueline M Miller¹² for the COVE Study Group****

[Line 17]: ****The members of the COVE Study Group are listed in the Supplementary Appendix*

Contributions

[Lines 341–344]: *“The trial sponsor, Moderna, Inc., was responsible for conceptualization and overall trial design (in collaboration with the Biomedical Advanced Research and Development Authority [BARDA], NIAID, Coronavirus Vaccine Prevention Network, and study co-chairs), site selection and monitoring, data analysis, preparation of the manuscript, and decision to publish.”*

Supplementary Appendix

[Lines 107–117]: *Lists of the COVE Trial Study Group and Investigators and Study Teams (PubMed Listed, and Ordered Alphabetically by Institution Affiliation)*

Comment 4. How were participants recruited and randomised? What were the exclusion and inclusion criteria?

Response: Thank you for the comment. The details on study design, and participant enrollment criteria have been published previously.^{1, 2} The trial enrolled adults aged 18 years of age or older, in a medically stable condition and with no known history of SARS-CoV-2 infection, at 99 US sites. Inclusion and exclusion criteria and other details of the study design and analyses are provided in the protocol and SAP, posted online as Supplementary Information with this article.

Comment 5. Please add a description of the study design and the point estimates of effect sizes to the abstract.

Response: Thank you for the comment. We have included key point estimates of effect sizes to the abstract.

Revised text:

Abstract

[Lines 26–41]: *“Coronavirus Efficacy (COVE) was a 3-part, phase 3, observer-blind, randomized, placebo-controlled (Part A) and open-label (Parts B and C) trial. Vaccination with two injections of mRNA-1273 (100 µg) was shown to be safe and efficacious at preventing coronavirus disease 2019 (COVID-19) at completion of the blinded part of the COVE study (Part A). We present the final report of the longer-term safety and efficacy data of the primary vaccination series plus a 50-µg booster dose administered in Fall 2021 (Parts B and C). The booster safety profile was consistent with that reported for the primary series. Incidences of COVID-19 and severe COVID-19 were higher during the Omicron BA.1 than Delta variant waves, and boosting versus non-boosting was associated with a significant, 47.0% (95% CI: 39.0–53.9%) reduction of Omicron BA.1 COVID-19 incidence (24.6 [23.4–25.9] vs 46.4 [40.6–52.7]). In an exploratory Cox regression model adjusted for time-varying covariates, a longer median interval between primary vaccination and boosting for the mRNA-1273 (13 months) versus placebo-mRNA-1273 (8 months) arm was associated with a significantly lower, 24.0% (16.0%–32.0%) risk of COVID-19 during the Omicron BA.1 wave). Boosting elicited greater immune responses against ancestral SARS-CoV-2 than the primary series, irrespective of prior SARS-CoV-2 infection.”*

Comment 6. Figure 1 – not all participants are accounted for. Within the placebo arm 1725 started open label – 25 completed but only 1576 shown as discontinued so there are 124 unaccounted for. Placebo crossed over arm – 12554 received 2nd injection, 9952 entered part C but only 2185 who discontinued are accounted for.

Response: Thank you for bringing this to our attention. We have updated Figure 1 to include the missing details as above.

Comment 7. It would be useful to report the level of precision when reporting AEs in order to compare the groups.

Response: The number of unsolicited adverse events and percentages are summarized for the primary series and booster safety sets. Confidence intervals were not a planned summary statistics for safety analysis and are not presented.

Comment 8. Line 107 describes that 65.4% of participants experienced AEs before the study end, 23.6% were related to the vaccine – can the authors provide further detail of what these were?

Response: Up to the end of the study, 65.4% had unsolicited AEs, including 23.6% who had events that were considered by the investigator to be related to study vaccine. The majority of unsolicited AEs reported after 28 days were due to underlying illness or intercurrent infection or

injury. Four SAEs considered by the investigator to be related to vaccine occurred up to 28 days post-vaccination, including one event each of myocarditis, coronary arteriospasm, heart flutter, and erythema nodosum, and are described in the Results section (lines 96–106). No additional SAEs or deaths considered by the investigator to be related to vaccination occurred after 28 days. We have also added information about the COVID-19 deaths that occurred during the study. We have revised the text to clarify.

Revised text:

Results

[Lines 112–120]: *“Adverse events that occurred after the booster dose and up to study-end were reported in 65.4% of all participants, predominantly attributable to underlying illness or intercurrent infection or injury, including 23.6% events considered by the investigator to be related to the study vaccine (Table S12). No additional SAEs considered by the investigator to be related to study vaccination occurred after 28 days. A total of 50 (0.3%) fatal AEs occurred as of study-end; none were attributed to study vaccine. Four deaths were associated with COVID-19, two during Part B (COVID-19 and pneumonia bacterial, and acute respiratory failure) and two after booster in Part C (respiratory failure [updated to COVID-19 after database lock] and COVID-19).*

Comment 9. It is interesting that more severe infections were seen in white compared with individuals from other ethnicities – this is contrary to much of the published literature. Can the authors comment on this?

Response: The finding of numerically lower incidence rates of severe Covid-19 in individuals in communities of color and numerically higher incidences for white individuals compared with the overall cohort is interesting. However note that the CIs of the incidence rates for these subgroups severe COVID-19 were overlapping with those of the overall cohort and thus both groups are considered comparable. We further investigated the demographic data by age group among the boosted participants, and observed that a greater proportion of White participants compared with non-White participants were aged ≥ 65 years (35.8% vs 13.6%). Figure S4 shows that the ≥ 65 years age group had a numerically higher incidence rate of severe COVID-19 than the < 65 years age group compared with the overall cohort. The higher incidence rate among White participants may possibly be explained by confounding between age group and race/ethnicity group.

Comment 10. The CIs for hazards ratios described in lines 189-191 of mRNA-1273 included 1 and were very wide suggesting low precision and lack of statistical significance. It would be important to note this in lines 236-238 of the discussion that make reference to this result.

Response: Thank you for the comment. In the exploratory model, we agree that the similarly reduced COVID-19 risk among boosted individuals in the mRNA-1273 and placebo-mRNA-1273 arms during the Delta and Omicron waves, as shown by hazard ratios that included 1 for the pre-boost comparison, indicate that efficacy from the primary immunization had decreased to a similar level for both variants. Comparison of the arms for the post-boost periods showed non-

significant reductions for Delta (0.86, 95% CI: 0.44-1.67) and significant reductions for Omicron (0.76, 95%CI: 0.68-0.84). We have clarified this in the Results and Discussion sections.

Revised text:

Results

[Lines 203–2078]: *“This corresponded to a non-significant risk reduction of 14% against Delta and a significant risk reduction of 24% against Omicron for the mRNA-1273 versus the placebo-mRNA-1273 arm during longer dosing intervals between the second injection and the booster for mRNA-1273 (median, 13 months) than placebo-mRNA-1273 (median, 8 months).”*

Discussion

[Lines 253–256]: *“The exploratory model indicated a significant reduction of Omicron-associated COVID-19 (24%), with longer dosing intervals between primary vaccination and boosting for mRNA-1273 (13 months) versus placebo-mRNA-1273 (8 months),³⁴ however, a lower, non-significant reduction (14%) was observed for Delta-associated COVID-19.”*

Comment 11. It is interesting that booster had longer duration of effectiveness in >65 vs <65 group (lines 197-198), can the authors comment on this?

Response: This is an interesting finding; however, to clarify our data indicates that a more pronounced effect of a booster dose is observed in those >65 years than <65 years and does not indicate a longer duration of effectiveness in those >65 years, and instead, the waning pattern of the 2 curves in Figure S14 are similar. These results, Lines 197-198, just elaborated the numbers without implication of longer duration of effectiveness in those >65 years. In line with these results, we mention in the discussion that higher risk is associated with older individuals and that higher numbers of older individuals are more likely to be boosted being at greater risk of COVID-19. We also note that higher risk is associated with extended time intervals prior to boosting due to waning of primary vaccination-conferred immunity and although we did not perform this analysis, may have affected the age groups as older participants were more likely to be boosted sooner.

Comment 12. In line with the SAGER guidelines, please can the authors present the results by sex and report how sex was taken into account in the study design.

Response: We provide data for sex (male, female) as a subgroup in the analyses of both COVID-19 and severe COVID-19 incidence rates (Figure 2 and Figure S4) and describe the results in the main text of the manuscript. Sex was a prespecified subgroup in the protocol study design and SAP, and was self-reported by participants at enrollment.

Comment 13. Lines 222-223 - These findings support the need for boosting to increase protection against emerging variants. As the authors hypothesise that infections occurred due to immune escape variants, it would be worth mentioning that boosters need to be updated to include most recent variants as well.

Response: Thank you for the comment. We have accordingly updated the text in the Discussion section.

Revised text:

Discussion

[Lines 237–240]: *“The decline in effectiveness seen over time was likely due to increased immune escape of Omicron variants, given that antibody levels remained substantially enhanced after boosting. These findings suggest the need for updated boosters that are closely matched to the circulating strains to increase effectiveness against emerging variants.”*

Comment 14. A number of long sentences that may be easier to understand if broken up (ie lines 201-205, 241-245).

Response: We have accordingly revised the discussion text for clarity.

Revised text:

Discussion

[Lines 216–220]: *“Follow-up of the COVE trial through its study end of April 2023 showed that primary vaccination with two injections of 100- μ g mRNA-1273 induced immune responses against SARS-CoV-2 that persisted through 13 months. Subsequent boosting with a 50- μ g dose elicited an immune response superior to that of primary vaccination, which remained higher than after primary vaccination for ≥ 6 months.”*

[Lines 260–266]: *“The effectiveness of boosting against COVID-19 in the model analysis as well as nAb increases from baseline were greater among SARS-CoV-2-negative participants aged ≥ 65 years than those 18-65 years, consistent with age-group effects shown previously.^{1,44} It is worth noting that higher risk is associated with extended time intervals prior to boosting due to waning of primary vaccination-conferred immunity,⁴⁴⁻⁴⁸ and that higher numbers of older than younger individuals are likely to be boosted and may be associated with a greater risk of COVID-19 exposure.⁴⁴⁻⁴⁷”*

Comment 15. Figure 2 – define IR, how was “at risk of severe COVID-19” defined?

Response: Thank you for bringing it to our attention. We have defined IR and risk factors for severe COVID-19 in the footnotes to Figure 2.

Revised text:

[Lines 511–519]: ***“Fig. 2. COVID-19 events based on adjudication committee assessments starting 14 days after booster, across subgroups (Part C per-protocol set). CI, confidence interval; IR, incidence rate; N, number of participants at risk. Incidence is defined as the number of participants with an event starting 14 days after booster by the number of participants at risk at 14 days after booster and adjusted by person-months (total time at risk) in each treatment group. 1 month = 30.4375 days. White is defined as White and non-Hispanic, and communities of***

color includes all the others whose race or ethnicity is not unknown, unreported, or missing. Participants at risk for severe COVID-19 included those aged ≥ 65 years and < 65 years with at least one of the risk factors (chronic lung disease, significant cardiac disease, severe obesity, diabetes, liver disease, or HIV infection)”

Comment 16. Table 5 – variation in decimal places reported

Response: Thank you for bringing this to our attention. We have formatted Table 5 for consistency in decimal points.

References

1. Baden LR, *et al.* Efficacy and safety of the mRNA-1273 SARS-CoV-2 vaccine. *N Engl J Med* **384**, 403-416 (2021).
2. El Sahly HM, *et al.* Efficacy of the mRNA-1273 SARS-CoV-2 vaccine at completion of blinded phase. *N Engl J Med* **385**, 1774-1785 (2021).
3. US Food and Drug Administration. Development and Licensure of Vaccines to Prevent COVID-19 Guidance for Industry. Preprint at <https://www.fda.gov/media/139638/download> (October, 2023).

REVIEWERS' COMMENTS

Reviewer #3 (Remarks to the Author):

Thank you for making the suggested changes which have addressed my concerns.